# The Role of Low-Molecular-Weight Organic Acids in Metal Homeostasis in Plants

**DOI:** 10.3390/ijms25179542

**Published:** 2024-09-02

**Authors:** Ilya V. Seregin, Anna D. Kozhevnikova

**Affiliations:** K.A. Timiryazev Institute of Plant Physiology, Russian Academy of Sciences, Botanicheskaya st., 35, Moscow 127276, Russia; ilya_seregin@mail.ru

**Keywords:** aluminum, cadmium, citrate, copper, iron, malate, manganese, metal accumulation in plants, metal detoxification, metal homeostasis, metal transport, nickel, organic acids, oxalate, tolerance, stress, zinc

## Abstract

Low-molecular-weight organic acids (LMWOAs) are essential O-containing metal-binding ligands involved in maintaining metal homeostasis, various metabolic processes, and plant responses to biotic and abiotic stress. Malate, citrate, and oxalate play a crucial role in metal detoxification and transport throughout the plant. This review provides a comparative analysis of the accumulation of LMWOAs in excluders, which store metals mainly in roots, and hyperaccumulators, which accumulate metals mainly in shoots. Modern concepts of the mechanisms of LMWOA secretion by the roots of excluders and hyperaccumulators are summarized, and the formation of various metal complexes with LMWOAs in the vacuole and conducting tissues, playing an important role in the mechanisms of metal detoxification and transport, is discussed. Molecular mechanisms of transport of LMWOAs and their complexes with metals across cell membranes are reviewed. It is discussed whether different endogenous levels of LMWOAs in plants determine their metal tolerance. While playing an important role in maintaining metal homeostasis, LMWOAs apparently make a minor contribution to the mechanisms of metal hyperaccumulation, which is associated mainly with root exudates increasing metal bioavailability and enhanced xylem loading of LMWOAs. The studies of metal-binding compounds may also contribute to the development of approaches used in biofortification, phytoremediation, and phytomining.

## 1. Introduction

Metals and metalloids are widely distributed in the environment, and many of them are essential for the functioning of all living organisms. Various human activities, including mining and mineral processing, agricultural activities, chemical production, metal smelting, and combustion of liquid and solid fuels, lead to a constant intensive flow of metals and metalloids into the environment, which makes the problem of pollution increasingly urgent [1,2]. Metals and metalloids, which enter the soil as a result of natural processes and anthropogenic activities, do not decompose and, therefore, are absorbed by plant root systems, making plants the main source of their entry into the food chains [3].

Arsenic (As) [4], lead (Pb), cadmium (Cd) [5,6], zinc (Zn) [7,8], nickel (Ni) [9,10], aluminum (Al) [11,12], and other elements [13,14] may exert multiple toxic effects on various physiological processes when supplied above certain thresholds. This leads to impaired growth and morphogenesis and reduced plant productivity (reviewed in [15]). Therefore, it has been proposed to call even the elements that are essential for plants, such as iron (Fe), cobalt (Co), manganese (Mn), copper (Cu), Zn, and Ni [16], as well as the elements necessary for some plant species, e.g., Al for *Camellia sinensis* [17], potentially toxic [16].

The metal accumulation capacity of different plant species and populations can vary significantly, which is largely determined by the different types of soils on which they had evolved [18,19,20,21,22,23]. Since metals enter plants mainly via the root systems, metals usually accumulate predominantly in the roots, while their transport into the shoots is restricted. Plant species with such metal accumulation pattern are considered excluders [24]. However, there are plant species that are metal-hypertolerant and capable of accumulating large amounts of metals in their shoots [25,26,27]. In 1976, such species were first named (hyper)accumulators in a work by Jaffré and co-authors describing *Pycnandra acuminata* (*Sebertia acuminata*), in which the Ni content in shoots was by 2–3 orders of magnitude higher than that in ordinary plants [28]. This term was later formulated based on the analysis of more than 2000 samples and 232 plant species from Ni-enriched soils in the work of Brooks and co-authors in 1977 [24]. There are certain threshold values of metal content in the shoots (mg kg^−1^ dry weight) of plants growing under natural conditions, above which a species can be classified as a hyperaccumulator [25,26,27]. Recently, a statistically derived approach showed that the historical hyperaccumulator thresholds are acceptably conservative [29]. To date, about 800 species of hyperaccumulators have been identified, most of which are Ni hyperaccumulators [26]. A few plant species, including *C. sinensis*, *Fagopyrum esculentum*, *Hydrangea macrophylla*, and *Melastoma malabathricum*, are capable of translocating and accumulating Al in the shoots up to concentrations above 1000 mg kg^−1^ dry weight, sometimes exceeding 3000 mg kg^−1^ dry weight, without symptoms of toxicity, and plant growth can even be stimulated by Al [30,31,32,33].

New species of hyperaccumulators are constantly being described, since this group of plants attracts close attention of scientists from various fields: biologists, ecologists, and geochemists. A novel approach for identifying new hyperaccumulator species involves X-ray fluorescence herbarium scanning [34,35]. Some of these species are considered as promising candidates for the development of phytoremediation [36] and phytomining technologies [37,38], aimed at decontamination and/or recultivation of contaminated areas or extracting metal from the above-ground plant organs and its further use for the production of metal-containing chemicals, respectively.

Plants use various strategies aimed at detoxifying metals and reducing the manifestation of their toxic effects: sequestration, exclusion, and chelation [14]. Low-molecular-weight ligands capable of forming stable complexes with metals take part in maintaining the labile pool of metals [39,40]. S-containing ligands (e.g., glutathione and phytochelatins) and O-containing ligands (e.g., organic acids (OAs)) are involved in metal transport and detoxification, while N-/O-containing ligands (e.g., nicotianamine and histidine) are also involved in hyperaccumulation mechanisms (reviewed in [39,40,41,42]). According to hard–soft acid–base theory, hard metal cations (e.g., magnesium (Mg^2+^), calcium (Ca^2+^), Co^3+^, Fe^3+^, and Al^3+^) form the most stable compounds with hard ligands (e.g., alcohols, amines, carboxylates, phosphates, and sulfates), in contrast to soft metal cations (e.g., mercury (Hg^2+^), Cu^+^, Cd^2+^, and Pb^2+^), which strongly bind to the soft ligands (e.g., thiols and phenyl groups) [43]. The discovery of nicotianamine-like metallophores in bacteria indicates an ancient emergence of the mechanisms involved in the tolerance to metals that were abundant in the early history of the Earth [44]. Malate, citrate, and oxalate play an important role in the detoxification of metals [45,46]. In general, the terms “malate”, “citrate”, “oxalate”, etc., denote the conjugate base of malic, citric, oxalic, etc., acids, respectively, but are widely used by researchers to refer to all physiological forms of these organic acids in plants [47] and will also be used by us further in this review for convenience.

In addition to participating in the maintenance of metal homeostasis, low-molecular-weight organic acids (LMWOAs) perform a number of essential functions in plants, participating in a variety of metabolic processes in cells, as well as in plant responses to biotic and abiotic stresses. Organic acids play a key role in energy metabolism, light-independent photosynthetic reactions, stomatal functioning, and glyoxylate pathways. They are the precursors for amino acid biosynthesis and modulate plant adaptation to the environment [48,49,50,51]. Since OAs are charged molecules, they are potential candidates for maintaining the pH in the cell and in the rhizosphere, electrical, and redox balance, as well as osmotic potential in cellular compartments [46,51,52,53,54,55,56]. Organic acids can accumulate in fruit, and their amount varies depending on the stage of fruit development and ripening [57]. Malate and citrate, which usually predominate in fruit [58,59], have a strong influence on organoleptic fruit quality [50,58,60]. Excess Ca in plants growing on alkaline soils is deposited in the vacuoles of root and shoot cells in the form of Ca oxalate [50]. Oxalate is involved in resistance to biotic stress [61], and acetate is involved in drought tolerance [62]. The drought-induced increase in acetate accumulation in *Arabidopsis thaliana* stimulates jasmonate-mediated signaling, which is an important component of the plant response to moisture deficiency [62]. Organic acids are involved in plant responses not only to drought stress, providing osmotic stress tolerance [63,64], but also to salt stress [52,54] and alkali stress [54,55]. Under phosphorus deficiency, the secretion of OAs into the rhizosphere contributes to extensive nutrient mobilization [64,65,66,67] and affects primary root growth [46]. OAs released into the rhizosphere contribute to a better availability of potassium in soil for absorption by plants [68]. Organic acids can act as chemoattractants for microbial root colonization [69,70,71,72] and as a source of carbon and energy for soil bacteria and fungi [73], which is important for plant–microbial interactions.

The essentiality of LMWOAs for plants as well as their participation in the transport, accumulation, and detoxification of metals has determined the relevance of this review. We apologize in advance to all authors whose work was not cited due to space limitations.

## 2. The Biosynthesis of LMWOAs

The main site of biosynthesis of malate, citrate, and oxalate is the mitochondria, where the reactions of the Krebs cycle, also known as the citric acid cycle or the di- and tri-carboxylic acid cycle, take place. In addition, OAs are formed in the glyoxylate cycle in the glyoxysomes, as a result of citrate catabolism and decarboxylation of malate and oxaloacetate in the cytosol, as well as in the processes of C4 and CAM photosynthesis (reviewed in [48,49,51,58,74]; Figure 1). Secondary reactions associated with the main metabolic pathways, such as the Krebs cycle, lead to the formation of other OAs, which is described in detail in the following reviews [51,75]. The biosynthesis of oxalate may be advantageous, since it is easier to metabolically engineer, as it is not an intermediate metabolite like citrate and malate. Under stressful conditions, an increase in the formation of OAs may occur due to (i) the conversion of phosphoenolpyruvate to oxaloacetate by phosphoenolpyruvate carboxylase (PEPC, EC 4.1.1.31) in the cytosol, (ii) the conversion of pyruvate to malate by the malic enzyme in the mitochondria, and (iii) the conversion of oxaloacetate to malate by malate dehydrogenase (MDH, EC 1.1.1.37) in the mitochondria [46].

## 3. The Contents of LMWOAs in Hyperaccumulator and Excluder Plants

Organic acids are among the most important metal-binding ligands in plants, with malate or citrate usually being the major ligand, although other LMWOAs may predominate in certain plant species. Malate plays an important role in binding Cd in the stems and leaves of the Zn/Cd hyperaccumulator *Sedum alfredii* [76] and in the leaves of the Cd hyperaccumulator *Solanum nigrum* [77], Ni in the leaves of the hyperaccumulators *Psychotria gabriellae* [78], *Leptoplax emarginata*, and *Odontarrhena muralis* (*Alyssum murale*) [79], as well as Zn in the shoots of the Zn hyperaccumulator *Arabidopsis halleri* [80,81]. At the same time, oxalate predominated in the leaves of the closely related excluder *Arabidopsis lyrata* when the plants were grown in the presence of Zn [81]. Aluminum was also bound to oxalate in the leaves of the Al accumulator *M. malabathricum* [30]. In the roots and shoots of the Ni hyperaccumulator *Brackenridgea palustris* ssp. *foxworthyi*, oxalate was assumed to act as a ligand chelating Ni and other metals [82]. Citrate was the predominant Ni-binding ligand in the latex of *P. acuminata*, stems of *L. emarginata* and *O. muralis*, as well as in the leaves of the hyperaccumulators from the genera *Homalium* and *Hybanthus* [28,79,83,84,85,86]. In addition, Ni citrate was found in significant quantities in the hyperaccumulator *Noccaea goesingensis* and the closely related excluder *Thlaspi arvense* [87]. Under Cd treatment, acetate predominated in the leaves of the Cd hyperaccumulator *Rorippa globosa* and the closely related Cd accumulator *Rorippa islandica* [88]. In the hyperaccumulator *S. nigrum,* the content of malate and citrate was higher than in the low-Cd-accumulating relative *Solanum torvum* [89]. In the metallophytes *Persicaria capitata*, *Persicaria puncata*, and *Conyza cordata*, Co tartrate predominated in all tissues, and Co citrate was present to a lesser extent in the leaves of *P. puncata* [21].

The contribution of different OAs to metal binding may differ not only between plant species, but also for different metals within the same species. For example, in the leaves of the Zn/Ni/Cd hyperaccumulator *Noccaea caerulescens*, which is considered as a model object for studying various aspects of metal hyperaccumulation [90], Cd was bound predominantly to malate [91], whereas Zn was bound mainly to citrate [92,93]. Hairy roots of *N. caerulescens* constitutively contained high levels of citrate, malate, and malonate [94]. Fumarate, *cis*-aconitate, and *trans*-aconitate were present in the shoots of *N. caerulescens* in small quantities, whereas formate and acetate were detectable only at high Zn concentrations [95].

Under the influence of various unfavorable factors, the contents of OAs in plant cells can change significantly. The LMWOA concentrations in roots usually range within 10 to 20 mM, and their concentrations in the cytosol range from 0.5 to 10 mM, which is about 1000-fold higher than that in the soil solution (0.5–50 µM). Moreover, they can increase manifold, even by an order of magnitude, under nutrient deficiency or an increased metal concentration exceeding the toxicity threshold [88,96,97]. A positive correlation was found between the accumulation of OAs and the concentration of metals in plants [98].

The influence of metals can be both species- and organ-specific, as well as metal-specific. For example, Cu treatment (100 μM) led to increased contents of succinate and tartrate in mature leaves of *Zea mays*, whereas Cd treatment (100 μM) led to increased contents of citrate and malate. Additionally, excess Cd or Cu induced an increase in the concentrations of tartrate and malate in roots, while Cu (100 μM) also peaked the citrate level [99]. In the leaves of *R. globosa*, a significant increase in the contents of water-soluble acetate, tartrate, and malate with the Cd concentration in the medium was observed, whereas in *R. islandica*, under the same conditions, only elevated content of acetate was detected [88]. Zinc, but not Cd, stimulated the synthesis of citrate in the roots of *N. caerulescens* [100,101]. In different plant organs, the LMWOA content may vary differently, which may be due to the uneven distribution of metals in plant tissues and organs. For example, under the effect of Cd, the contents of OAs decreased in the shoots and increased in the roots of *Dittrichia viscosa*, which is consistent with the significantly higher metal contents in the roots [102]. The contents of citrate, malate, oxalate, and *cis*-aconitate in the shoots increased under the influence of Ni and were higher in *Lolium perenne* than in *Z. mays*, while more malate, oxalate, and *cis*-aconitate accumulated in the roots of the latter (cited from [45]).

One of the reasons for the changes in LMWOA content in metal-treated plants may be metal-induced oxidative stress and changes in the cytosolic pH, which are accompanied by an increase in the activity of the Krebs cycle enzymes and in the formation of LMWOAs. For example, under the influence of Al, an increase in the activity of citrate synthase (CS, EC 4.1.3.7), MDH, PEPC [103,104], and NADP-malic enzyme (NADP-ME, EC 1.1.1.40) [105] was observed, as well as a decrease in the activity of aconitase (ACO, EC 4.2.1.3) [104,106]. In *Saccharum officinarum*, Al-induced enhancement of the expression of the *SoCYS* and *SoMDH* genes encoding CS and MDH was observed, which confirmed the participation of citrate and malate in plant response to Al [107]. At the same time, in Al-treated *Secale cereale* and *Senna tora* (previously, *Cassia tora*), there was an increase only in CS activity, but not in MDH and PEPC activities, with an increase in the secretion of malate/citrate and citrate, respectively [106,108], which indicates the complexity of the relationship between the activity of various enzymes and the biosynthesis of OAs.

Higher levels of activity of OA metabolism enzymes may contribute to plant tolerance. For example, the Al-tolerant *Phaseolus vulgaris* genotype Quimbaya had higher specific activities of CS (4-fold) and PEPC (1.6-fold) compared to those in the Al-sensitive genotype VAX-1 under Al stress [109]. In the Mn-tolerant genotype of *Stylosanthes guianensis*, the production of malate increased in response to Mn, which was accompanied by an increase in the expression level of *SgMDH* [110].

However, despite the fact that the hyperaccumulators *A. halleri* [80,81,111], *N. caerulescens* [95,100,101,112], and *R. globosa* [88] constitutively contain large amounts of malate, LMWOAs are mainly involved in metal detoxification rather than in hyperaccumulation mechanisms, which is determined by the lower stability of their complexes with metals compared to other metal-binding ligands (reviewed in [40,113,114,115]). In addition, in the hyperaccumulators *R. globosa* and *A. halleri*, the total content of OAs in leaves was even lower than in closely related non-hyperaccumulating species [81,88], which confirms the above statement. In general, metal-induced changes in the contents of OAs allow plants to adapt to metal excess in the environment via the formation of various complexes of OAs with metals.

## 4. Complexes of LMWOAs with Metals

The negatively charged carboxyl groups of OAs allow them to bind cations to form salts or complexes (Figure 2). 

Strong bonds can be formed by chelating metals with the carboxyl groups of citrate, malate, malonate, oxalate, tartrate, or other LMWOAs, which act as electron donors [98]. In aqueous solution, Cu(II) and Co(II) ions tend to form monomers and dimers with citrate, which was confirmed by electron paramagnetic resonance spectroscopy, whereas Ni(II) forms only monomers [118]. Using hydrophilic interaction chromatography mass spectrometry (HILIC-MS) and size exclusion chromatography electrospray ionization mass spectrometry (SEC-ESI-MS), Fe(III)-(Citrate)_2_, Fe(III)-(Malate)_2_, Fe(III)_3_-(Malate)_2_-(Citrate)_2_, Fe(III)_3_-(Malate)_1_-(Citrate)_3_, Mn-(Citrate)_2_, Zn-(Citrate)_2_, Ca-(Citrate)_2_, and Mg-(Citrate)_2_ were detected in the xylem sap and embryo sac liquid of *Pisum sativum*. Fe(II)-(Malate)_2_, Mn-(Malate)_2_, Co(II)-(Citrate)_2_, Ni-(Malate)_2_, and Ni-(Citrate)_2_ were identified only in the xylem sap, while Fe(III)_3_-(Malate)_3_-(Citrate)_1_ and Fe(II)-(Citrate)_2_ were found only in the embryo sac liquid [119]. In the xylem sap of *Solanum lycopersicum*, Fe(III)_2_-(Citrate)_2_ or Fe(III)_3_-(Citrate)_3_ complexes prevailed depending on the ratio of Fe to citrate [120]. In the leaves of *M. malabathricum*, Al-(Oxalate)_1_, Al-(Oxalate)_2_, and Al-(Oxalate)_3_ complexes were identified [30]. In the apoplast, Fe was present in the form of complexes with citrate and malate (Fe(III)_3_-(Citrate)_2_-(Malate)_2_, Fe_3_(III)-(Citrate)_3_-(Malate)_1_, and Fe(III)-(Citrate)_2_) [121,122]. Since the precise structure of the resulting compound is rarely determined in the majority of studies, further in the text we will use the term “complex” for all compounds of OAs with metals.

The stability of metal ion complexes with S-, N-, and O-containing ligands changes according to the Irving–Williams series: Zn^2+^ < Cu^+^ > Cu^2+^ > Ni^2+^ > Co^2+^ > Fe^2+^ > Mn^2+^ > Mg^2+^ > Ca^2+^, where at equimolar concentrations, each previous ion can replace the subsequent one at the binding sites [123,124]. However, in biological systems in vivo, the situation is always more complicated due to different (sometimes by orders of magnitude) concentrations of metals in the cell cytoplasm, as well as due to different contents of various metal-binding ligands that have different affinities for metal ions [40,41,42,125].

The stability of the resulting complexes of OAs with metals is determined by their stability constants. Logarithms of stoichiometric stability constants (log *K*) for 1:1 complexes of metal ions with citrate or malate are, respectively: Fe^3+^ (11.0/7.1), Al^3+^ (7.9/6.0), Cu^2+^ (5.9/4.2), Ni^2+^ (5.4/5.2), Ca^2+^ (4.9/2.7), Zn^2+^ (4.8/2.9), Co^2+^ (4.7/3.1), Fe^2+^ (4.4/2.5), Pb^2+^ (4.0/2.4), Mn^2+^ (3.7/2.2), and Cd^2+^ (3.8/1.9) [96,126]. The stability of metal complexes with OAs depends on the metal:ligand ratio in the complexes. For example, the values of stability constants for 1:1 and 1:2 complexes of metal ions and oxalate are, respectively: Fe^3+^ (7.7/13.6), Al^3+^ (6.1/11.1), Cu^2+^ (6.2/4.0), and Mn^2+^ (3.9/4.4) [96]. Based on the given values of stability constants, it is clear that trivalent metal ions have a higher affinity for OAs, and they are mobilized (and immobilized) most readily by OAs. The stability of complexes of a particular metal with different OAs also varies significantly. For example, the pK values of stability constants of Cd-acetate, Cd-lactate, Cd-glycolate, Cd-maleate, Cd-succinate, and Cd-citrate were 1.5, 1.7, 1.9, 2.4, 2.1, and 3.8, respectively [127]. In general, the mono-OAs (e.g., acetate^−^) have lower metal-complexing ability and chelate far fewer cations than tri-OAs (e.g., citrate^3−^) and di-OAs (e.g., malate^2−^ and oxalate^2−^) [96,128,129]. Therefore, the latter play a more important role in maintaining metal homeostasis. It has been shown that the stability constants of complexes of OAs with metals are significantly lower than those of metal complexes with such ligands as histidine and the non-proteinogenic amino acid nicotianamine [126,130]. Therefore, all things being equal, metals will form more stable complexes with the latter. However, we still know very little about in vivo competition between different ligands for the same metal and between different metals for the same ligand.

The efficiency of binding of various ligands to metals and the stability of the resulting complexes is also determined by the ratio of protonated and deprotonated ligand forms, which depends on the pH (reviewed in [40,41,42]; Table 1 and Figure 2). 

At physiological pH values, OAs are present in the form of anions and can form complexes with cations, and the stability of these complexes varies for different OAs depending on the pH [46,96]. The stability of metal complexes with OAs increases with the number of carboxyl groups. The chelating capacity of citrate is significantly higher than that of malate due to the presence of three carboxyl groups in citrate but two carboxyl groups in malate [96,128]. However, the cellular pool of malate is often higher than that of citrate, and in some tissues reaches 10 mM [132,133]. It is obvious that OAs can bind metals at neutral pH values of the cytosol (pH 7.2–7.5) and phloem sap (pH 7–8), but due to the higher values of the stability constants of metal complexes with histidine and nicotianamine, metals will form complexes predominantly with the latter. At the acidic pH values of the vacuolar sap (pH 4.5–6) and the xylem sap (pH 5–6.2), at which the stability of complexes with histidine and nicotianamine is low [40,42], OAs will play a key role in metal binding. The concentration of OAs in the xylem sap and vacuolar sap can reach millimolar values, which is quite sufficient for the formation of complexes with metals [119]. Therefore, LMWOAs may play an important role in the detoxification of metals in the vacuole, as well as in their long-distance transport via the xylem. Acidic pH of the apoplast also favors the complexation of metals with carboxylic acids [120]. Besides the pH value of the surrounding medium, which is regarded as the dominant factor of ligand exchange, in case of trace metals with multiple oxidation states, which act as electron carriers (Fe^2+^/Fe^3+^ and Cu^+^/Cu^2+^), the ligand switch was also suggested to be determined by the adjacent redox reaction [43].

As a result of binding of Cd, Cu, Fe(II), and Pb with the carboxyl groups of OAs, in particular those of citrate, mononuclear tridentate complexes are formed, which are thermodynamically stable and are not biodegraded, while bidentate complexes of citrate with Fe(III) and Ni are rapidly degraded [134,135]. However, the possibility of transport of OA complexes with metals across biological membranes remains poorly studied. It has been proposed that Cd is translocated in the form of organometallic complexes in soil solution toward the root surface, and that these complexes can disassociate to release free Cd^2+^ at the root surface to be absorbed by the roots [136]. The possibility of the transport of Al complexes with malate across the plasma membrane has been shown [137]. However, it is obvious that the transport of metal complexes with OAs requires further research, which is problematic due to the complexity of their identification.

In general, plants use two important OA-mediated mechanisms aimed at reducing the toxic effects of metals, which include the secretion of OAs into the rhizosphere to prevent the entry of metals into the roots, as well as the sequestration of metals in the vacuoles of root or shoot cells [32,40,138,139]. Let us consider these mechanisms in more detail.

## 5. Secretion of LMWOAs 

### 5.1. Sources of LMWOAs in Soil

In the solutions of the upper soil layers of forest ecosystems, citrate, malate, and oxalate predominate among the aliphatic LMWOAs, while substituted and unsubstituted OAs, such as benzoate and cinnamate, predominate among the aromatic LMWOAs. The main sources of the aliphatic and aromatic LMWOAs in soils are root exudates, plant residues, and microbial metabolites [140]. Therefore, the balance of LMWOAs in the rhizosphere is determined by their secretion by plant roots, microbial mineralization of organic compounds, and sorption–desorption processes [96].

### 5.2. Composition of Root Exudates

Root exudates are fluids containing organic and inorganic compounds secreted by the roots into the rhizosphere during plant growth and development [141]. Plants secrete a plethora of primary and secondary metabolites into the rhizosphere in order to facilitate interactions with their biotic and abiotic environment. In addition to inorganic ions (Cl^−^, SO_4_^2−^, CO_3_^2−^, PO_4_^3−^, and NH_4_^+^), root exudates contain low-molecular-weight compounds (e.g., sugars, carboxylates, amino acids (including phytosiderophores), as well as flavonoids, coumarin, sorgoleone, etc.) and high-molecular-weight compounds (e.g., proteins, such as exoenzymes) [46,64,141,142,143,144,145,146]. The composition of root exudates may vary under various mineral deficiencies or metal excess (reviewed in [46]). Graminaceous species engage a specific method of enhanced Fe uptake, known as Strategy II, involving phytosiderophores (phytometallophores), which are secreted into the rhizosphere and can effectively bind and increase the bioavailability of not only Fe, but also that of Cd, Co, Cu, Ni, Mn, and Zn. Phytosiderophores include 2′-deoxymugineic acid as well as various compounds of the mugineic acid family [40,42]. Mugineic acid derivatives were found not only in the root exudates of cereals, but also in the root exudates of some dicotyledonous plant species [147,148], which indicates a wider distribution of these compounds in nature. Moreover, the hyperaccumulator *A. halleri* is capable of secreting nicotianamine as a phytosiderophore, which resembles Strategy II, although this species belongs to the dicotyledonous family Brassicaceae [149]. At the same time, OAs are major components of root exudates [46,64,144]. Among the LMWOAs found in root exudates there are acetate, ascorbate, citrate, maleate, malate, fumarate, lactate, oxalate, succinate, tartrate, and some other OAs (Table 2).

**Table 2 ijms-25-09542-t002:** Low-molecular-weight organic acid secretion by the roots of different plant species under metal(-loid) treatment.

Species	Element Concentration	Growth Medium	Duration of Exposure	Organic Acids Secreted	References
**Acanthaceae**					
*Avicennia marina*	2–30 µmol L^−1^ As	Sand culture irrigated with nutrient solution	3 months	Citrate, malate, oxalate	[150]
**Amaranthaceae**					
*Amaranthus* sp.	50 µM Al	0.5 mM CaCl_2_	3 h	Oxalate	[151]
*Amaranthus hypochondriacus*	25 µM Al	0.5 mM CaCl_2_	0.5–9 h	Citrate, oxalate	[152]
	10–50 µM Al		6 h		
*Halimione* *portulacoides*	67 µg L^−1^, 6.9 mg L^−1^ Cu	Solution	2 h	Oxalate	[153]
	13 µg L^−1^, 1.1 mg L^−1^ Cd				
*Spinacia oleraceae*	50 µM Al	0.5 mM CaCl_2_	3 h	Oxalate	[151]
	10–100 µM Al	0.5 mM CaCl_2_	6 h	Oxalate	[154]
**Araceae**					
*Colocasia esculenta*	900 µM Al	Nutrient solution	10 days	Oxalate	[155]
**Asteraceae**					
*Cichorium intybus*	0.4, 0.8, 1.6 mg kg^−1^ Cd	Soil	60 days	Acetate, fumarate, malate, oxalate	[136]
*Helianthus annuus*	1, 5, 10 mg kg^−1^ Cd	Soil	50 days	Acetate, malate, maleate, succinate	[156]
	0.125–2.0 mg L^−1^ Cu	Nutrient solution	10 days	Citrate	[157]
	5, 20 g m^−3^ Al	Soil	3, 10 days	Citrate, fumarate, malate	[158]
	5, 20 g m^−3^ Zn		10 days	Citrate, fumarate, malate	
	5 g m^−3^ Zn		3 days	Fumarate, malate	
	20 g m^−3^ Zn			Citrate, fumarate, malate	
	5, 20 g m^−3^ Cd		3, 10 days	Fumarate	
**Brassicaceae**					
*Arabidopsis thaliana*	2.7 µM Al	Nutrient solution	24 h	Citrate, malate, pyruvate, succinate	[159]
	50 µM Al	Nutrient solution	48 h	Citrate, malate	[160,161]
	1.5 µM Al	Nutrient solution	48 h	Citrate, malate	[162]
	700 µM Al	Agar plates	3 d	Citrate, fumarate, α-ketoglutarate, lactate, malate, succinate	[163]
	10 µM Al	0.5 mM CaCl_2_	6 h	Citrate	[164]
	10 µM Al	Nutrient solution	24 h	Malate	[165]
	500 µM Al or 500 µM Ga	Nutrient solution	2 d	Citrate, malate	[166]
	10 µM Al	Root exudation collection medium	24 h, 2–12 h	Malate	[167]
	0.4 µM Cd		24 h		
	2 µM Cu				
	1.3 µM Er				
	1.3 µM La				
*Arabis alpina*	4.59 mg kg^−1^ Cd and 392 mg kg^−1^ Pb	Soil	120 days	Citrate, glyoxylate, malate, oxalate, tartrate	[168]
*Brassica campestris* ssp. *chinensis Makino*	13, 52 mg L^−1^ Zn	Nutrient solution	12 days	Acetate, citrate, lactate, malate, oxalate, succinate, tartrate	[169]
*Brassica napus*	50 µM Al	0.5 mM CaCl_2_	6 h every day for 10 days (intermittent treatment)	Citrate, malate	[170]
	50 µM Al	0.5 mM CaCl_2_	6 h, 5–15 h, and 10 days of intermittent treatment	Citrate, malate	[103]
*Brassica oleraceae*	50 µM Al	0.5 mM CaCl_2_	1–24 h	Citrate	[171]
	50 µM Al	0.5 mM CaCl_2_	1–12 h	Malate	[172]
*Camelina sativa*	25 µM Al	Nutrient solution	2–12h	Malate	[173]
*Raphanus sativus*	50 µM Al	0.5 mM CaCl_2_	6 h every day for 10 days (intermittent treatment)	Citrate	[170]
**Caryophyllaceae**					
*Viscaria vulgaris*	25, 75 µM Al	Nutrient solution	24 h	Citrate, oxalate	[174]
**Crassulaceae**					
*Sedum alfredii*	10, 20, 40, 80 mg L^−1^ Zn; 1, 2, 4, 8 mg L^−1^ Cd	Nutrient solution	4 weeks	Oxalate, tartrate	[175]
	1000 mg L^−1^ Zn (different Zn salts)	Nutrient solution	15 days	Acetate, citrate, formate, malate, mesylate, succinate (depending on Zn salt)	[176]
	5, 50, 100 µM Cd	Nutrient solution	1 month	Citrate, malate, oxalate, succinate, tartrate	[177]
	5, 10, 40, 400 µM Cd	Nutrient solution	4 days	Fumarate, 2-hydroxyacetate, lactate, oxalate, succinate	[178]
	5 µmol L^−1^ Cd	Nutrient solution	4 days	Fumarate, lactate, oxalate, succinate	[179]
			8 days	2-Hydroxyacetate, lactate, oxalate, succinate	
	10 µmol L^−1^ Cd		4 days	Lactate, succinate	
			8 days	Lactate	
	10, 50, 200, 1000 µmol L^−1^ Pb	Nutrient solution	4 days	Citrate, glycerate, lactate, oxalate, succinate, galactonic acid	[180]
	40 µM Cd	Nutrient solution	4, 8 days	Lactate, oxalate, succinate	[181]
	25 µM Cd	Nutrient solution	3 days	Citrate, malate, oxalate, tartrate	[182]
	16.5 mg kg^−1^ Cd	Soil	56 days	Malate, oxalate, tartrate in hyperaccumulating ecotype	[183]
				Malate, oxalate in non-hyperaccumulating ecotype	
*Sedum plumbizincicola*	40, 400 µmol L^−1^ Cd	Nutrient solution	7 days	Citrate, glutarate, glycolate, 4-hydroxybutyrate, levulinate, malate, succinate	[184]
**Cyperaceae**					
*Carex pilulifera*	25, 75 µM Al	Nutrient solution	24 h	Citrate, oxalate	[174]
	41 or 63 µM Al (quickly reacting) in soil, followed by 90 µM Al in the nutrient solution	Soil, then nutrient solution	6 months in soil, 3h in the nutrient solution	Acetate, citrate, formate, lactate, malate, oxalate, succinate	
**Fabaceae**					
*Acacia auriculiformis*	160 mM, 950 mM Al^3+^	0.2 mM CaCl_2_	24 h	Citrate, oxalate	[185]
*Acacia mangium*	5 mM Al	Nutrient solution	28 d	Citrate, malate	[186]
*Glycine max*	10, 30, 50 or 70 µM Al	0.5 mM CaCl_2_	24 h	Citrate	[187]
	50 µM Al		2–12 h		
	1 mM Al	Nutrient solution	0, 2, 4, 6, 8, 10, 12, 14 days	Citrate, malate, oxalate	[66]
	38 µM Al	4.3 mM CaCl_2_	6–72 h	Citrate, malate, oxalate	[188]
	10–70 µM Al	Solution	4 h	Citrate	[189]
	30 µM Al		0.5–4 h		
	38 µM Al	4.3 mM CaCl_2_	6 h	Malate	[190]
	25, 50, 200 µM Al	0.5 mM CaCl_2_	24 h	Citrate	[191]
	30 µM Al	0.5 mM CaCl_2_	9 h	Citrate	[104]
	30 µM Al	0.5 mM CaCl_2_	2–24 h	Citrate	[105]
*Leucaena leucocephala*	5 mM Al	Nutrient solution	28 d	Citrate, malate, succinate	[186]
*Lupinus albus*	50 µM Al	0.5 mM CaCl_2_	10–180 min, 12 h	Citrate	[192]
	0.125–2.0 mg L^−1^ Cu	Nutrient solution	10 days	Citrate	[157]
*Medicago sativa*	5 µM Al	0.5 mM CaCl_2_	2–24 h	Malate	[193]
* Paraserianthes falcataria * (*Falcataria falcata*)	5 mM Al	Nutrient solution	28 d	Citrate, malate, succinate	[186]
*Phaseolus vulgaris*	148 µM Al	Nutrient solution	8 d	Citrate	[194]
*Pisum sativum*	20 µM Al	0.2 mM CaCl_2_	12 h	Citrate	[195]
*Senna (Cassia) tora*	50 µM Al	0.5 mM CaCl_2_	2–12 h	Citrate	[196]
	10–50 µM Al	0.5 mM CaCl_2_	9 h		
	20 µM Al	0.2 mM CaCl_2_	12 h	Citrate	[195]
	20, 50 µM Al	0.5 mM CaCl_2_	3–12 h	Citrate	[106]
	100 µM Al	0.5 mM CaCl_2_	3–12 h	Citrate	[197]
	20 µM Al	0.5 mM CaCl_2_	12 h	Citrate, malate, oxalate	[198]
*Stylosanthes guianensis*	50 µM Al	0.5 mM CaCl_2_	3 h	Citrate	[199]
	10–30 µM Al		24 h		
*Stylosanthes scabra*	50 µM Al	0.5 mM CaCl_2_	3 h	Citrate	[199]
	10-30 µM Al		24 h		
*Vicia faba*	4.59 mg kg^−1^ Cd and 392 mg kg^−1^ Pb	Soil	120 days	Citrate, glyoxylate, malate, oxalate, tartrate	[168]
*Vigna mungo*	50 µM Al	Solution	48 h	Citrate, malate	[200]
*Vigna umbellata*	50 µM Al	0.5 mM CaCl_2_	3–12 h	Citrate	[201]
	25 µM Al	Nutrient solution	3–24 h	Citrate	[202]
	5–50 µM Al		24 h		
	25 µM Al	0.5 mM CaCl_2_	3–12 h	Citrate	[203]
	25 µM Al	0.5 mM CaCl_2_	0.5–9 h	Citrate	[164]
**Malvaceae**					
*Gossypium hirsutum*	20 µM Al	Solution	48 h	Citrate	[204]
**Myrtaceae**					
*Eucalyptus camaldulensis*	160 mM Al^3+^	0.2 mM CaCl_2_	24 h	Oxalate	[185]
	950 mM Al^3+^			Citrate, oxalate	
	25–75 µM Al	Nutrient solution	24 h	Citrate, malate	[205]
	1 µM Cu			Malate	
*Melaleuca cajuputi*	160, 950 mM Al^3+^	0.2 mM CaCl_2_	24 h	Citrate, oxalate	[185]
*Melaleuca leucadendra*	160, 950 mM Al^3+^	0.2 mM CaCl_2_	24 h	Citrate, oxalate	[185]
**Nephrolepidaceae**					
*Nephrolepis exaltata*	67, 267 µM As	Nutrient solution	2 days	Oxalate, phytate	[206]
**Onagraceae**					
*Oenothera picensis*	0.125–2.0 mg L^−1^ Cu	Nutrient solution	10 days	Citrate, fumarate, succinate	[157]
**Plantaginaceae**					
*Plantago lanceolata*	0.4, 0.8, 1.6 mg kg^−1^ Cd	Soil	60 days	Fumarate, malate, oxalate	[136]
*Veronica officinalis*	25, 75 µM Al	Nutrient solution	24 h	Citrate, oxalate	[174]
**Poaceae**					
*Avena sativa*	50 µM Al	0.5 mM CaCl_2_	6 h every day for 10 days (intermittent treatment)	Citrate	[170]
*Brachiaria brizantha*	20 µM Al	0.2 mM CaCl_2_	12 h	Citrate	[195]
*Brachiaria decumbens*	43, 115 {Al^3+^}	Nutrient solution	13 days	Citrate, malate, oxalate	[207]
*Brachiaria ruziziensis*					
*Brachypodium distachyon*	20 µM Al	Nutrient solution	3 h, 24 h	Citrate, malate	[208]
*Deschampsia flexuosa*	25, 75 µM Al	Nutrient solution	24 h	Citrate, oxalate	[174]
	41 or 63 µM Al (quickly reacting) in soil, followed by 90 µM Al in the nutrient solution	Soil, then nutrient solution	6 months in soil, 3 h in the nutrient solution	Acetate, citrate, formate, lactate, malate, oxalate, succinate	
*Festuca gigantea*	25, 75 µM Al	Nutrient solution	24 h	Citrate, oxalate	[174]
*Holcus lanatus*	50 µM Al	0.5 mM CaCl_2_	3, 6, 9, 12, 24 h	Citrate, malate	[209]
	25, 50 or 100 µM Al		24 h		
*Holcus mollis*	25, 75 µM Al	Nutrient solution	24 h	Citrate, oxalate	[174]
*Hordeum vulgare*	20 µM Al	0.2 mM CaCl_2_	12 h	Citrate	[195]
	10 µM Al	1 mM CaCl_2_	6 h	Citrate	[210]
*Imperata condensata*	0.125–2.0 mg L^−1^ Cu	Nutrient solution	10 days	Citrate, oxalate	[157]
	0.125 mg L^−1^ Cu			Succinate	
*Lolium perenne*	10, 20, 50, 100 µM Mn	Nutrient solution	15 days	Citrate, malate, oxalate, succinate	[211]
*Milium e* *ffusum*	25, 75 µM Al	Nutrient solution	24 h	Citrate, oxalate	[174]
*Miscanthus floridulus*	10, 50, 100, 200 µM Cd	0.5 mM CaCl_2_	24 h	Malate	[212]
*Miscanthus sacchariflorus*	100 µM Cd		4 or 24 h		
*Oryza glaberrima*	20 µM Al	0.2 mM CaCl_2_	12 h	Citrate	[195]
*Oryza sativa*	20 µM Pb	Pb(NO_3_)_2_ solution or distilled water	6 d	Oxalate	[213]
	10, 50 mg kg^−1^ Cd	Soil	40 d	Acetate, citrate, formate, malate, oxalate, tartrate	[214]
	46 µg g^−1^ total Hg, 3.7 ng g^−1^ MeHg	Soil	70 days	Citrate, lactate, malate, oxalate, succinate	[215]
	25 µM Tl	Tl(NO_3_)_3_ solution or deionized water	12 h	Oxalate	[216]
	2, 5 mg L^−1^ Cd	Nutrient solution	not mentioned	Acetate, citrate, malate, malonate, oxalate, succinate, tartrate	[217]
	20 µM Al	0.2 mM CaCl_2_	12 h	Citrate	[195]
	50, 100 µM Cr	Nutrient solution	8, 16 d	Acetate, citrate, lactate, oxalate, malate, succinate	[218]
	50 µM Cd, 200 µM Zn separately and in combination, as well as with 1.5 mM K_2_SiO_3_	Nutrient solution	7 days	Acetate, fumarate, maleate, oxalate, tartrate	[219]
	25 µM Tl (III)	3 mM Ca(NO_3_)_2_ and 1 mM MgSO_4_	12 h	Oxalate	[216]
*Phragmites australis*	67 µg L^−1^ Cu	Solution	2 h	Citrate, maleate, oxalate	[153]
	6.9 mg L^−1^ Cu			Citrate, maleate	
	29 µg L^−1^ Ni			Citrate	
	5 mg L^−1^ Ni			Citrate, oxalate	
	13 µg L^−1^ Cd			Citrate, oxalate	
	1.1 mg L^−1^ Cd			Citrate, maleate, oxalate	
*Phyllostachys pubescens*	200 µM Pb	Nutrient solution	5 days	Malate, oxalate	[220]
	100 µM Zn				
	25 µM Cu			Lactate, malate, oxalate	
	10 µM Cd				
*Saccharum officinarum*	2.10 mM Al (505.9 Al^3+^ free activity)	Nutrient solution	12 d	Citrate, malate	[107]
*Secale cereale*	50 µM Al	0.5 mM CaCl_2_	2–12 h	Citrate, malate	[108]
	10–50 µM Al		24 h		
*Setaria italica*	50–300 mg kg^−1^ Cd	Soil	3 weeks	Acetate, butyrate, lactate, malate, propionate, succinate	[221]
*Sorghum bicolor*	20 µM Al	0.2 mM CaCl_2_	12 h	Citrate	[195]
	27 µM Al^3+^	Nutrient solution	1, 3, 6 d	Citrate	[222]
*Sorghum bicolor* × *Sorghum sudanense*	0.5, 5 mg L^−1^ Cd	0.5 mM CaCl_2_	5 h	Malate	[223]
*× Triticosecale* Wittmack	50 µM Al	0.5 mM CaCl_2_	6 h	Citrate, malate	[224]
*Triticum aestivum*	50 µM Al	Nutrient solution	24 h	Malate, oxalate, succinate	[225]
	200 µM Al	0.2 mM CaCl_2_	80 min	Malate	[226]
	50 µM Al	0.2 mM CaCl_2_	2–25 h	Citrate, malate	[227]
	50 µM Al	0.5 mM CaCl_2_	6 h every day for 10 days (intermittent treatment)	Malate	[170]
	50 µM Al	0.5 mM CaCl_2_	2–12 h	Citrate, malate	[108]
	10–50 µM Al		24 h		
	25–100 µM Al	0.5 mM CaCl_2_	24 h	Malate	[228]
	100, 200 µM Al	Nutrient solution	9 days	Citrate, malate	[229]
	20 µM Al	0.2 mM CaCl_2_	12 h	Citrate, malate	[195]
	10 µM Al	0.5 mM CaCl_2_	12 h	Citrate, malate, oxalate	[198]
*Triticum turgidum* var. *durum*	100–400 µg kg^−1^ total Cd	Soil	2 weeks	Acetate, butyrate, citrate, fumarate, malate, oxalate, propionate, succinate, tartrate	[230]
*Zea mays*	10–40 µM Cd	Sand culture irrigated with nutrient solution	6 weeks	Aconitate, cinnamate, citrate, glycerate, lactate, propionate	[145]
	20 µM Al	0.2 mM CaCl_2_	12 h	Citrate, malate	[195]
	6 µM Al	Nutrient solution	24–48 h	Citrate	[231]
	9 µM Al		6–18 h	Citrate, malate	
	10–50 µM Al		20 h		
	5–80 µM Al^3+^	4.3 mM CaCl_2_ solution/nutrient solution	1–4 d	Citrate	[232]
	3.8, 10, 20, 30, 40, 50 µmol kg^−1^ Cd	Soil	48 h	Acetate, citrate, formate, malate, oxalate, succinate	[233]
	0.5, 5 mg L^−1^Cd	0.5 mM CaCl_2_	5 h	Citrate	[223]
	100 µM Cd or Cu	Nutrient solution	4 days	Citrate, lactate, malate, succinate, tartrate	[234]
	1000 mg L^−1^ Zn (different Zn salts)	Nutrient solution	15 days	Acetate, citrate, formate, mesylate, succinate, tartrate (depending on Zn salt)	[176]
**Polygonaceae**					
*Fagopyrum esculentum*	50 µM Al	0.5 mM CaCl_2_	3 h	Oxalate	[151]
	50 µM Al	0.5 mM CaCl_2_	6 h every day for 10 days (intermittent treatment)	Oxalate	[170]
	50 µM Al	0.5 mM CaCl_2_	6 h	Oxalate	[235]
	150 µM Al		3 h		
	100 µM Al	0.5 mM CaCl_2_	3–12 h	Oxalate	[197]
	50 µM Al	0.5 mM CaCl_2_	12 h	Citrate, malate, oxalate	[198]
	10 µM Al	0.5 mM CaCl_2_	24 h	Citrate, malate	[236]
	10–50 µM Al		6 h	Citrate	
	30 µM Al		3–12 h		
*Rumex acetosella*	25, 75 µM Al	Nutrient solution	24 h	Citrate, oxalate	[174]
**Pteridaceae**					
*Pteris vittata*	67, 267, 1068 µM As	Nutrient solution	2 days	Oxalate, phytate	[206]
**Rhizophoraceae**					
*Kandelia candel*	5–50 ppm Cd	Sediment	6 months	Acetate, butyrate, citrate, formate, fumarate, lactate, malate, maleate, tartrate	[237]
*Kandelia obovata*	2.5–40 mg L^−1^ Cd	Sand culture irrigated with nutrient solution	1–24 h, 3, 7, and 14 days	Acetate, formate, citrate, fumarate, lactate, malate, maleate, oxalate, succinate, tartrate	[238]
**Rosaceae**					
*Geum urbanum*	25, 75 µM Al	Nutrient solution	24 h	Citrate, oxalate	[174]
**Rubiaceae**					
*Galium saxatile*	25, 75 µM Al	Nutrient solution	24 h	Citrate, oxalate	[174]
**Rutaceae**					
*Citrus grandis*	1 mM Al	Sand culture irrigated with nutrient solution	18 weeks	Citrate, malate	[239]
	500 µM Al	0.5 mM CaCl_2_	12 h, 24 h		
*Citrus sinensis*	1 mM Al	Sand culture irrigated with nutrient solution	18 weeks	Citrate, malate	[239]
	500 µM Al	0.5 mM CaCl_2_	12 h, 24 h		
	500 µM Al	0.5 mM CaCl_2_	24 h	Citrate	[240]
			12 h	Malate	
	1 mM Al for 18 weeks, followed by 24 h or 12 h of 500 µM Al	Sand culture irrigated with nutrient solution (18 weeks), then 0.5 mM CaCl_2_	18 weeks and 24 h	Citrate	
			18 weeks and 12 h	Malate	
**Salicaceae**					
*Populus trichocarpa*	500 µM Al	0.5 mM CaCl_2_	3–12 h	Citrate	[241]
**Solanaceae**					
*Capsicum annuum*	2, 10 µM Cd	Nutrient solution	21 days	Acetate, citrate, oxalate, succinate, tartrate	[242]
*Nicotiana benthamiana*	1, 5, 10 mg kg^−1^ Cd	Soil	50 days	Acetate, glycolate, lactate, maleate, succinate	[156]
*Nicotiana tabacum*	0.5 and 1 mg L^−1^ Cd	Nutrient solution	6 days	Acetate, formate, lactate, malate, maleate, oxalate, propionate, succinate, tartrate	[243]
*Solanum lycopersicum* (*Lycopersicon esculentum*)	100–300 mg L^−1^ Cr	Nutrient solution	1 week	Acetate, citrate, maleate, oxalate, tartrate	[244]
	50 µM Al	0.5 mM CaCl_2_	3 h	Oxalate	[151]
	10 µM Cd	0.5 mM CaCl_2_	0.5–4 h	Oxalate	[245]
	10–50 µM Cd	0.5 mM CaCl_2_	1 h		
	1, 5, 10, 20 mg kg^−1^ Cd	Soil	5 weeks	Acetate, citrate, malate, oxalate, tartrate	[246]
*Solanum nigrum*	1, 5, 10, 20 mg kg^−1^ Cd	Soil	5 weeks	Acetate, citrate, malate, oxalate, tartrate	[246]
	1, 5, 10, 20 µM Cd	Nutrient solution	1 week	Acetate, citrate, malate, tartrate	[247]
**Theaceae**					
*Thea sinensis*	20 µM Al	0.2 mM CaCl_2_	12 h	Citrate	[195]
**Ulmaceae**					
*Ulmus laevis*	0.06, 0.6 mM NaAsO_2_, Na_2_HAsO_4*_7H_2_O or dimethylarsinic acid (single and in different combinations)	Sand culture irrigated with nutrient solution	3 months	Acetate, citrate, formate, fumarate, malate, malonate, oxalate, succinate (depending on the As form, its concentration, and combination)	[248]

The contents of LMWOAs in the rhizospheric soil solutions can vary widely: from micromolar (10 μmol L^−1^) to millimolar (5 mmol L^−1^) concentrations [39,131,249]. Depending on the plant species, microbial activity, soil type, nutrient supply, and other factors, the quantitative and qualitative LMWOA composition of root exudates may vary greatly [66,96,250,251] (Table 2). For example, in the root exudate of *A. thaliana*, citrate and malate dominated, and it also contained fumarate, ketoglutarate, lactate, pyruvate, and succinate [159,160,161,162,163]. In the root exudate of *Nicotiana tabacum*, oxalate and succinate predominated, while acetate, formate, lactate, and maleate were present in smaller quantities [243]. In *Vitis amurensis*, the root exudate predominately contained oxalate [55], whereas in *Oryza sativa* it contained acetate, malate, oxalate, succinate, tartrate, and other OAs [213,214,215,216,252]. In mangrove plants, *Kandelia candel*, the main component of root exudates was citrate [237], whereas oxalate was the major LMWOA in the exudates of *Kandelia obovata* [238,253], *Avicennia marina* [150], and *Aegiceras corniculatum* [253]. In addition, even in different varieties of the same species, the composition of root exudates may vary. The concentration of oxalate in the root exudates of two *Z. mays* cultivars grown under the same conditions differed two-fold, while the concentration of tartrate differed by almost an order of magnitude [254]. Significant varietal differences in the composition of root exudates were also revealed in *O. sativa* [214,215,217,252]. All these facts complicate comparing the results of experiments conducted on different species and under different growth conditions.

The quantitative and qualitative composition of root exudates can change under the influence of metals, and the amount of LMWOAs often increases. Thus, under Cd treatment, an increase in the contents of OAs in root exudates was observed in the high-Cd-accumulating line and the normal line of *O. sativa*, but Cd had no effect on the composition of root exudates of the two rice lines [217]. The two cultivars of *Capsicum annuum* differing in their ability to accumulate Cd in fruit did not differ in the total content of OAs in the exudate. However, the composition of LMWOAs depended on the cultivar under study and Cd exposure levels [242]. In plants of the two lines of *N. tabacum* contrastingly different in their capacity to accumulate Cd, various Cd-induced changes were observed in both the quantitative and qualitative composition of LMWOAs in root exudates [243]. High Cd levels in the medium stimulated the secretion of LMWOAs, and their total concentrations in root exudates were significantly correlated with the amount of Cd accumulated in shoots and roots of *Setaria italica* [221]. The increase in the contents of OAs, mainly citrate and aconitate, was also observed in the root exudates of Cd-treated *Z. mays*, while the contents of amino acids and sugars decreased, and the contents of proteins did not change [145]. In two cultivars of *Solanum lycopersicum* treated with chromium (Cr), an increase in the contents of acetate, citrate, maleate, tartrate, and oxalate was observed [244]. The ratio of LMWOAs in the exudate largely depended on soil type [230], as well as on the metal concentration in the medium, as, for example, was shown for Cd-treated *K. obovata* and *K. candel* [237,238] and As-treated *A. marina* [150].

Metal-induced changes in the composition of root exudates may differ in plant species and varieties that are metal-tolerant and metal-sensitive. For example, Cd treatment significantly reduced the contents of most alkaloids, OAs, and phenolic acids in the exudates of the Cd-tolerant cultivar of *Panicum miliaceum*, while it increased the contents of most lipid fatty acids and phenolic acids in the exudates of the Cd-sensitive cultivar. Moreover, in both varieties, Cd treatment significantly increased the contents of *trans*-aconitate in the exudates, the role of which in Cd tolerance is not clear yet [255]. Obviously, all these changes can be considered as a plant response aimed at adapting to unfavorable environmental factors.

### 5.3. Functions of Root Exudates

Root exudates perform a variety of functions related not only to the mineral nutrition of plants, but also to plant–plant, plant–microbe, and plant–insect interactions (reviewed in [46,64,141,142]). Organic acids, along with phenols and flavonoids from root exudates, play a key role in plant–microbe symbiosis. For example, malate was shown to be a carbon source and chemoattractant for rhizobacteria [69], and its secretion by *O. sativa* roots was stimulated by plant-growth-promoting rhizobacteria *Bacillus subtilis* [256]. Organic acid secretion is also a widespread plant response to alkalinity [55].

Organic acids found in the rhizosphere affect metal solubility, mobility, and phytoavailability in the soil [140,143,218], which is important both under metal deficiency and metal excess in the environment. Plant adaptation to a lack of nutrients includes changes in the architecture of the root system, changes in the pH of the soil solution, and the release of root exudates, which promotes solubilization of the minerals in the rhizosphere [257,258]. Deficiency of various elements, such as Fe [259], Mn [260], and Zn [261], leads to increased secretion of OAs. Organic acids form complexes with ions in the soil, which is accompanied by the release of protons, leading to a decrease in the pH of the soil solution [262] and an increase in metal solubility and mobility in the vicinity of the roots, and thereby enhance metal phytoavailability [144,146,246]. In addition, OAs can stimulate the growth of beneficial microorganisms that increase the availability of metals for plant roots [46], whereas the formation of stable complexes of LMWOAs with metals can limit the entry of metals into the plant and the manifestation of their toxic effects, which allows plants to cope with metal-induced stress (see below). Thus, despite significant losses of carbon due to OA secretion, OAs play a greater role in the absorption of mineral elements compared to amino acids, which are also constituents of root exudates [46], and the participation in maintaining metal homeostasis is an important function of OAs.

### 5.4. The Role of LMWOA Secretion in Metal Detoxification

The beginning of the studies on the participation of LMWOAs in metal uptake by plants is historically associated with the studies of the mechanisms of Al detoxification. At low-pH conditions (below 5.5), the ionic forms of Al (Al^3+^ or Al(H_2_O)_6_^3+^), a natural constituent of the clay fraction of the soil, are highly phytotoxic, damaging the roots and hampering plant development [11,12,32,263]. The root tip, especially the transition zone located between the elongation zone and the apical meristem, is the most sensitive target for Al toxic effects [208,263,264,265,266]. In 1986, Kitagawa drew attention to Al-induced malate secretion by *Triticum aestivum* roots [267]. In 1991, Miyasaka and co-authors showed that the plants of the Al-tolerant variety of *P. vulgaris* secreted much more citrate than the plants of the Al-sensitive variety [194]. However, the most convincing evidence of the important role of OAs in metal tolerance was obtained from a pair of near-isogenic lines of *T. aestivum* (T3 and ES3), differing in tolerance to Al [225,268]. Under Al treatment, the secretion of malate occurred quite quickly, which allowed the researchers to conclude that a previously existing mechanism had been activated [225]. It was shown that Al-induced secretion of OAs occurred primarily in the apical root zone [31,188,190,192,222,225,226,235,263,269], where the signs of Al toxicity were the most pronounced. Al-induced release of OAs into the apoplast can prevent metal binding to the pectins of the cell wall matrix [263]. Limiting the release of OAs to the apical root zone might reduce the metabolic cost and carbon loss for Al detoxification while maximizing the protection of the root cells which are the most susceptible to Al-induced damage [31,266].

Under Al treatment, the secretion of oxalate, malate, and/or citrate was shown depending on the species under study (Table 2). The species-specific pattern of Al-induced OA secretion was confirmed by the experiments conducted on different plant species under the same growth conditions: under the treatment with 50 µM Al, *F. esculentum* secreted mainly oxalate, *T. aestivum* secreted malate, whereas *A. sativa*, *B. napus*, and *R. sativus* secreted both citrate and malate [170]. The types of OAs secreted by different plant species may depend on the abundance of specific membrane-localized transporters and anion channels being responsible for OA secretion [129]. In general, citrate, oxalate, and malate are the major OAs secreted by plant roots in response to Al treatment, with their affinity for Al^3+^ ions decreasing in the order from citrate to malate [198].

The data on the dependence of the intensity of LMWOA secretion and/or plant tolerance to Al on the endogenous contents of LMWOAs and the activity of OA metabolism enzymes in roots remain controversial. On the one hand, there are a number of studies in which such a dependence was not found [151,193,197,224,225,231,269,270]. On the other hand, in Al-treated roots of *C. sinensis*, malate (citrate) secretion was positively correlated with the malate (citrate) level [240]. In the roots of Al-tolerant *F. falcata*, the Al-induced accumulation and release of citrate were associated with the enhanced activity of mitochondrial CS (mCS) due to the Al-inducible expression of *mCS* [186]. The Al-triggered biosynthesis and release of citrate have been shown for various Al-tolerant plants, including *B. napus* [103], *S. tora* [106], *G. max* [104], and *S. cereale* [108]. Obviously, just increasing the internal OA level is not enough for their increased secretion since in any case, some transport processes must be to a certain degree involved in the Al-induced secretion of OA anions [269], which requires their thorough study for understanding the reasons for the observed discrepancies. In this regard, the studies conducted using the *nadp-me1* mutant of *A. thaliana* lacking NADP-ME1, which catalyzes the oxidative decarboxylation of malate and shows strong expression in the root apex, are of interest [271]. The mutant plants and the wild-type plants exuded similar amounts of malate in response to Al, had similar levels of expression of the gene encoding the malate-transporting protein, and accumulated similar amounts of Al, suggesting that the higher tolerance of *nadp-me1* to Al is not associated with the mechanism of Al exclusion. The higher Al tolerance of *nadp-me1* seems to result not only from the increase in the levels of malate in roots but also from a modification of the signaling pathways [271].

In early studies, two distinct patterns of OA release were identified depending on its rapidity [31,196]. Plant species engaging pattern I, such as *Hordeum vulgare* [210], *T. aestivum* [108,225,272], and *F. esculentum* [197], can rapidly release citrate, malate, or oxalate, respectively, under Al stress, which apparently involves the activation of pre-existing transporters [210,272]. Plant species engaging pattern II, such as *S. tora* [196,197] and *G. max* [187], intensively secrete citrate after at least 4 h of Al treatment, whereas for *Holcus lanatus*, it takes 6 h to start the release of malate from the roots [209]. This is consistent with the data on time-dependent OA secretion for other species [108,167,173,199,202,203,208,222,231,241], suggesting that in this case, gene induction is required [31]. The causes and pathways for the emergence of two such distinct OA secretion patterns are not completely clear from an evolutionary point of view.

The intensity of OA secretion by plants can vary over time and shows intraspecific differences. *Vigna umbellata* and *P. trichocarpa* initially responded to Al treatment by secreting small amounts of citrate, while after a longer exposure, large amounts of citrate were released, which is aimed at minimizing OA anion release in order to optimize root carbon use efficiency with respect to the alleviation of Al toxicity [164,241]. Biphasic secretion of first malate and then citrate by roots was found in *S. cereale* [108], which is determined by the involvement of various channels/transporters in this process [108,164]. In *G. max*, similar amounts of citrate were secreted by two Al-treated cultivars (Jiyu 70 and Jiyu 62) before Al treatment. After 12 h of Al treatment, Jiyu 70 secreted more citrate than Jiyu 62, and after 24 h, a sustained increase in internal malate and citrate concentrations was observed in Jiyu 70, whereas in Jiyu 62, the citrate content decreased, which is consistent with the greater tolerance of the former [105]. Additionally, it was supported that Al-induced citrate efflux depended on the malate pool in *G. max* root apices [105,187].

Higher Al tolerance is often related to higher rates of OA exudation by plants, which leads to the formation of stable, non-toxic complexes of Al with OAs in the rhizosphere [107,159,161,173,194,204,208,225,226,231,235,239,272]. Thus, the secretion of LMWOAs in the root apex is a widespread mechanism aimed at binding Al in the apoplast and rhizosphere, limiting its entry into the roots and reducing the manifestation of its toxic effects. In this case, rhizodermal cells should be the most tolerant to metal [200,208,239,269,273]. However, in a number of cases, no clear dependence of plant tolerance on root OA exudation was found [151,195,207,232], suggesting that for some plant species the effect of OAs is masked by some other mechanisms, and/or that the Al-induced secretion of OAs is sometimes insufficient to be an effective detoxification mechanism (reviewed in [269]).

Similarly to Al, species-specific exudation of LMWOAs plays an important role in the detoxification of As, Cd, Cr, Cu, Mn, Ni, Pb, Zn, mercury (Hg), gallium (Ga), and thallium (Tl), resulting in reduced bioavailability of these metals/metalloids and alleviation of their toxic effects (Table 2). Intriguingly, the Ga-triggered responses in *A. thaliana* are similar to those triggered by Al. Similarly to Al, in *A. thaliana*, Ga is detoxified externally by citrate and malate secretion, potentially via the formation of non-toxic Ga(III)-OA complexes in the rhizosphere or apoplastic space [166]. It was also suggested that the secretion of oxalate and citrate by Mn-tolerant *L. perenne* cultivars decreased Mn bioavailability in the rhizosphere, thus contributing to the Mn tolerance of these cultivars [211]. In the roots and root exudates of Pb-tolerant *O. sativa* varieties, a Pb-stimulated increase in oxalate content was observed, whereas the opposite was shown for the sensitive varieties [213]. Unlike *Helianthus annuus* and *L. albus*, which secreted citrate at constant levels, or the Cu-metallophyte *Imperata condensata*, which showed Cu-induced citrate exudation, in the Cu-metallophyte *Oenothera picensis*, an extremely high content of succinate was found in the root exudate of Cu-treated plants. This is unusual for plants due to the low stability of Cu complexes with succinate, which is approximately 1100 times lower compared to that for the Cu complexes with citrate [157]. Treatment of two-year-old *Ulmus laevis* with As(III) or As(V) individually or in a mixture led to an increase in the concentrations of LMWOAs in the rhizosphere, especially those of oxalate and malonate, in comparison to the control, whereas in the roots, there was a decrease in the overall content of the profiled LMWOAs, which indicates the participation of OA exudation in plant tolerance to As [248]. Generally, Cd-tolerant *Miscanthus sacchariflorus* exhibited higher malate exudation rates than Cd-sensitive *Miscanthus floridulus* under Cd stress [212]. In *S. lycopersicum*, Cd-induced oxalate secretion from the root apex promoted Cd exclusion from the roots, contributing to lower Cd accumulation in Cd-tolerant cultivars compared to Cd-sensitive cultivars, not only in the short-term hydroponic experiment but also in the long-term hydroponic and soil experiments [245].

The LMWOA efflux from plant roots in hydroponic solutions may differ from that in rhizosphere soils due to the differences in root morphology as well as microbial and nutrient status between the hydroponic and the soil environment. Al-induced secretion of LMWOAs was observed only in wild-growing *D. flexuosa*, *G. saxatile*, *R. acetosella*, *V. officinalis*, and *V. vulgaris*, but not in plants grown hydroponically [174]. It was reported that low-Cd cultivars of *Triticum turgidum* var. *durum* secreted more total LMWOAs than high-Cd cultivars in sterile nutrient solution cultures [274], but the opposite was the case in pot experiments when the plants were grown in the presence of Cd [230]. Alleviation of Cd toxicity and increased plant tolerance due to the release of root exudates has been shown for different plant species grown in hydroponics as well as in soil (Table 2).

Metal effects on the secretion of root exudates is not only species-specific but is also metal-specific. Under different Cd, Ni, and Cu treatments, *Phragmites australis* secreted oxalate, citrate, and maleate, while *Halimione portulacoides* exuded oxalate and maleate. Maleate was present in very low amounts in the root exudates of Cu-treated *P. australis* and *H. portulacoides* exposed to non-contaminated medium. At the same time, Cu affected the exudation of oxalate by *H. portulacoides* and that of oxalate and citrate by *P. australis*, while Ni and Cd did not stimulate any specific response [153]. Experiments with *H. annuus* under Al, Cd, and Zn treatment revealed a prominent increase in the contents of malate and citrate in the roots and shoots, hence enabling the plants to tolerate metal-induced stress. However, the secretion of these OAs in Cd-treated seedlings was negligible [158]. Under Tl stress, only oxalate was the specific OA in the root exudates of *O. sativa* [216]. Acetate, formate [214], or tartrate [217] were found to be the major OAs secreted by the roots of Cd-treated *O. sativa*, whereas oxalate, citrate, and malate were predominant under Cr treatment [218], and malate was predominant in Hg-treated cultivars [215], which, apparently, is also a consequence of the existence of clear intraspecific differences in metal-induced secretion in some plant species.

In addition to citrate, oxalate, and malate, other LMWOAs found in root exudates may be involved in metal detoxification. Moreover, the contents of different LMWOAs can vary differently depending on the plant species and metal content in the environment. Oxalate, fumarate, malate, and acetate secreted by *Cichorium intybus*, as well as oxalate, fumarate, and malate secreted by *Plantago lanceolata*, were the major LMWOAs in plants grown in soil-filled rhizocolumns under increasing Cd levels (0, 0.4, 0.8, and 1.6 mg Cd kg^−1^ soil) for 60 days. Compared with *P. lanceolata*, *C. intybus* secreted less fumarate and more acetate under all Cd treatments. The content of oxalate secreted by both species did not change with the increasing soil Cd levels, the content of acetate in the root exudate of *C. intybus* decreased by 50%, while the contents of malate and fumarate in the root exudate of this species did not change at 0.4 or 0.8 mg kg^−1^ Cd, but increased by 76% and 140%, respectively, at 1.6 mg kg^−1^ Cd, compared to the control [136]. Fumarate is a dicarboxylic acid and has the greatest affinity toward Cd^2+^ ions. The accumulation of fumarate, malate, oxalate, and succinate in the roots of low-Cd-accumulating isoline of *T. turgidum* (W9260-BC) was associated with more Cd being sequestered in roots, which prevented Cd translocation into the shoots [275]. The greater secretion of LMWOAs, including fumarate, by *G. max* (cultivar AC Hime) treated with 3.3 mg Cd L^−1^ reduced Cd bioavailability and uptake by plants compared to the control due to the formation of Cd-OA complexes in the soil (cited from [136]).

The formation of metal complexes with OAs in the rhizosphere can not only reduce but can also increase the accumulation of toxic elements in plants, which has been thoroughly studied in the case of Cd and shown for such excluder species as *H. annuus*, *Nicotiana benthamiana* [156], *S. lycopersicum* [246], and *O. sativa* [214]. However, the increase in Cd accumulation in some cases may not be accompanied by a decrease in plant tolerance to Cd [156] as a result of the lower toxicity of Cd complexes with organic ligands compared to that of the ionic form [40,41,146], as well as due to effective intracellular mechanisms of metal detoxification. It was shown that *O. sativa* cultivars characterized by high Hg accumulation secreted more OAs [215]. Exudation of malate was associated with Cd accumulation in *O. sativa* [217], whereas Cr accumulation was significantly and positively correlated with the exudation of oxalate, malate, and citrate [218]. Further studies of this issue are of practical importance for agriculture and the development of phytoremediation technologies.

Different LMWOAs may have different effects on metal uptake. Efficient Cd uptake and accumulation by plants is often associated with an increase in the content of acetate in the rhizosphere [136,214,230,276]. Acetate was found in the root exudates of *C. annuum* [242], *C. intybus* [136], *K. candel* [237], *K. obovata* [238], *O. sativa* [214], *S. lycopersicum*, *S. nigrum* [246], *Z. mays* [233], and some other species (Table 2). For example, Cieśliński and co-authors reported that the significant increase in acetate (by 163%) and other OAs in the rhizosphere of a high-Cd-accumulating cultivar of *Triticum durum* (Kyle), compared to a low-Cd-accumulating cultivar (Arcola), could explain the 33% greater total Cd accumulation in Kyle compared to Arcola from Sutherland sandy loam soil with a total Cd concentration of 0.41 mg kg^−1^ [230]. However, the mere presence of acetate in root exudates does not indicate an increase in Cd accumulation. The effect of root exudates on metal uptake by plants largely depends on the ratio of their components, including different LMWOAs, as well as on plant growth conditions.

### 5.5. Secretion of LMWOAs by Hyperaccumulators

Organic acids do not play a significant role in the mechanisms of hyperaccumulation due to the low stability constants of metal complexes with OAs [40,114,115]. However, the secretion of OAs may have a certain effect on the intensity of metal uptake, which can make some contribution to the process of hyperaccumulation [146]. Hyperaccumulators have a higher metal complexation and extraction capacity, releasing root exudates that increase the bioavailability of metals [277]. The presence of Cd in the medium led to an increase in the release of secondary metabolites into the rhizosphere by the hyperaccumulator *S. alfredii*. After 4 days of incubation in the presence of Cd (5–400 µM), 62 compounds were identified using gas chromatography-mass spectrometry (GC-MS), of which the contents of 20 compounds changed under the different Cd treatments [178]. Under different growth conditions and at different concentrations of Cd in the medium, the root exudates of *S. alfredii* contained citrate, fumarate, galactonate, glycerate, lactate, malate, oxalate, succinate, and tartrate, as well as oleic acid, tetradecanoic acid, threonic acid, and some other acids (Table 2) [175,176,177,178,179,180,181,182,183,278].

The chemical composition of the exudate varied not only depending on the growth conditions, but also depending on the *Rhizobium rhizogenes* strain present in the growth medium, which influenced the development of the root system, the accumulation of Cd and Zn, and the rate of metal phytoextraction [278]. Inoculation with different bacterial species may have diverse effects on the secretion of root exudates. Inoculation with the metal-tolerant bacterium *Burkholderia cepacia* reduced the secretion of OAs by *S. alfredii*, especially that of tartrate, which, however, was accompanied by an increase in the accumulation of Cd and Zn and plant tolerance to them [175]. At the same time, after the inoculation with the endophytic bacterium *Sphingomonas SaMR12*, an increase in the secretion of oxalate, citrate, and succinate and in the accumulation of Cd, as well as improved growth of *S. alfredii*, were observed [177]. Further studies of the effects of bacterial inoculation on the secretion of OAs by roots, metal uptake, and plant metal tolerance will be beneficial for the development of phytoextraction techniques.

The composition of root exudates may differ between the ecotypes of *S. alfredii*, pointing to the existence of intraspecific differences in the secretion of LMWOAs. Oxalate, malate, and tartrate were the predominant LMWOAs in the rhizosphere soil solution of the hyperaccumulating ecotype of *S. alfredii.* However, almost no tartrate was detected in the rhizosphere soil solution of the non-hyperaccumulating ecotype. Cadmium accumulation in the hyperaccumulating ecotype of *S. alfredii* was promoted by the exudation of tartrate, which was highly efficient in Cd solubilization due to the formation of soluble Cd-tartrate complexes [183]. In addition, oxalate secretion predominantly by the root apices of the Cd-tolerant ecotype of *S. alfredii* was two times higher than that of the non-tolerant ecotype [182], which may also contribute to Cd uptake by plants. Phenylglyoxal, an inhibitor of OA secretion inactivating the anion channels, effectively blocked Cd-induced oxalate and tartrate exudation, which was accompanied by a decrease in the Cd contents in the roots and shoots of the Cd-hyperaccumulating ecotype of *S. alfredii* [182,183].

In the root exudate of the closely related Cd hyperaccumulator *Sedum plumbizincicola*, 155 metabolites were detected, among which 33 showed significant differences in accumulation under Cd stress, including OAs, amino acids, lipids, and polyols. Cadmium suppressed OA metabolism and lipid metabolism in *S. plumbizincicola* and significantly affected amino acid metabolism. In particular, Cd inhibited the secretion of malate, glycolate, glutarate, and other OAs by roots [184]. The data obtained suggest that the quantitative and qualitative composition of root exudates can vary differently even in closely related hyperaccumulator species and, therefore, the role of LMWOAs in hyperaccumulation in different plant species may be intricate.

Currently, the data on the involvement of root exudates in metal uptake by hyperaccumulators from other genera are scarce (Table 2). It has been shown that acetate, citrate, malate, oxalate, and tartrate secreted by the roots of *S. nigrum* increased Cd accumulation and tolerance in plants under increased Cd levels [246,247]. As in *S. alfredii* [183], a positive correlation between the exudation of tartrate and the accumulation of Cd by plants was found in *S. nigrum* [247]. Under Fe deficiency, there was an increase in the exudation of LMWOAs and, similarly to the Fe-sufficient plants, a positive correlation between Cd accumulation and the release of malate and acetate was observed. In contrast to the Fe-sufficient plants, in the Fe-deficient plants, there was a negative correlation between Cd accumulation and the exudation of tartrate [247].

Root exudates of different hyperaccumulator species may have different effects on the solubility of various metals. The release of organic ligands contributed to increased availability of Ni in the rhizosphere of *N. goesingensis* [279], that of Cr in *Leersia hexandra* [280], and As in *Pteris vittata* [206,281], while the root exudates of *N. caerulescens* did not significantly increase the mobilization of Cd and Zn and, therefore, did not participate in the mechanisms of hyperaccumulation [282]. The influence of LMWOAs on the content of metals in plant organs can be determined not only by their involvement in metal uptake by root systems, but also by their role in metal transport and detoxification in plants.

## 6. LMWOA-Mediated Metal Detoxification in the Vacuole

Metals enter the vacuole in the ionic form or as complexes with organic ligands, e.g., phytochelatins, via various transporters (reviewed in [40,41,114,123,283,284,285,286,287,288]). Early studies showed that 79–90% of absorbed Cd was located in the cell wall [289], and nearly all Cd found in protoplasts was present in the vacuole [290]. In plants, the majority of OAs, such as malate and citrate, are typically stored in the vacuole, where they are involved in osmoregulation and function as counterions [291,292,293]. While the concentration of malate and citrate in the cytosol rarely exceeds 5 mM, their concentration in the vacuole can be several times higher [96]. In the roots and shoots of the hyperaccumulator *Noccaea praecox*, up to 80% of the Cd ligands were O-containing ligands provided by the cell walls and by the OAs stored in the vacuole [294]. Metal binding to OAs is especially important in the leaf epidermal cells, where vacuoles occupy up to 99% of the cell protoplast volume, which leads to restricted transport of metals to the mesophyll cells and thus diminishes their negative effects on photosynthesis (reviewed in [40,295]). In the vacuoles of leaf epidermal cells in hyperaccumulators, the concentration of metals can reach several hundred millimoles per liter [296,297], which determines the necessity for their detoxification via binding to OAs.

The possibility of transport of metal complexes with OAs across the tonoplast remains disputable. It is assumed that OAs are capable of binding metal ions in the cytosol, limiting their toxic effects and facilitating their entry into the vacuole in bound form [129,234], as, for example, in the case of Cd binding to malate, with subsequent transport into the vacuoles in *S. nigrum* leaf cells [77]. It has also been hypothesized that upon entering the vacuole, metal complexes with malate are destroyed, and metals bind to stronger ligands, such as citrate and oxalate, while malate is re-exported into the cytosol [45,234,298]. The possibility of transport of Ni citrate [299] and Zn citrate (Kozhevnikova et al., unpublished data) into the vacuole was shown on tonoplast vesicles isolated from the roots of the hyperaccumulator *N. caerulescens* and the excluder *T. arvense*. However, the cytosol contains ligands whose affinity for metal ions is much higher than those of OAs. The cytosolic pH (7.2–7.5), which is favorable for the formation of sufficiently strong metal complexes with nicotianamine, histidine, and phytochelatins, makes the mechanism of metal detoxification with the participation of OAs in the cytosol less significant compared to the one involving other ligands [40].

As in the cytosol, in the vacuoles of root and shoot cells, besides OAs, other metal-binding ligands are present, including nicotianamine, histidine, and phytochelatins (reviewed in [40,41,42]), indicating the possible importance of these compounds in metal detoxification. It is assumed that nicotianamine is capable of binding Ni and Zn in the vacuole, and at the pH of the vacuolar sap, the complexes of nicotianamine with Ni are more stable than those with Zn. However, compared with the stability of these complexes in the cytosol, their stability in the vacuolar sap at pH < 6 is significantly lower [300,301]. The possibility of transport of histidine complexes with Ni and Zn into the vacuole has been shown [299,302]. However, due to the lower pH of the vacuolar sap (4.5–6.0) compared to the pH of the cytosol (pH 7.2–7.5), the nitrogen of the imidazole ring of histidine is protonated and the stability of the metal-histidine complex decreases. Therefore, it can be assumed that this complex is destroyed in the vacuole [40]. Complexes of metals with phytochelatins that enter the vacuole can also probably be destroyed due to the more acidic pH of the vacuolar sap [41]. Considering the high contents of OAs in the vacuole, their contribution to metal binding in the vacuolar sap is notable, despite the fact that OAs are weaker ligands compared to the above-mentioned ones.

Transporters of the NRAMP (natural resistance-associated macrophage protein) family located at the tonoplast, in particular NRAMP3 and NRAMP4, are involved in the remobilization of Fe(II), Mn, Zn, Ni, and Cd from the vacuole into the cytosol, whereas COPT5, which belongs to the copper transporter (CTR/COPT) family, is involved in the remobilization of Cu (reviewed in [40,123,283,284,303]). Considering that metals in the vacuole are predominantly found in the form of complexes with OAs, it remains unclear how metals are delivered to the transporter to be subsequently transported in the ionic form into the cytosol. The relatively low stability of metal complexes with OAs, compared to that with other ligands, may be of some importance in this case. As a result, metal ions are transferred to the metal-binding domains of transporters containing amino acid residues that strongly bind metal ions. It is obvious that metal transport across the tonoplast is an important component of metal homeostasis.

## 7. The Role of LMWOAs in the Long-Distance Transport of Metals

Long-distance transport via the xylem is a key factor determining metal accumulation in the above-ground organs. Metal complexes with LMWOAs are one of the forms in which metals are transported from roots to shoots via the xylem, which ensures negligible adsorption of uncharged or negatively charged complexes by negatively charged xylem cell walls [304]. It is assumed that LMWOAs are the major metal-binding ligands in the xylem sap [40,305], which is determined by both low pH values (5–6.2) and high concentrations of OAs in the xylem sap. However, the amounts of metals bound to OAs may vary depending on plant species as well as metal species. Modeling of Ni speciation in the xylem sap of *Odontarrhena lesbiaca* (*Alyssum lesbiacum*) indicated that Ni was chelated mainly by histidine (19%), glutamine (15%), citrate (9%), and malate (3%), while up to 48% was predicted to be the free (hydrated) Ni ions [306]. Field studies have shown that in the xylem sap of the hyperaccumulator *Odontarrhena serpyllifolia* ssp. *lusitanica* (*Alyssum serpyllifolium* ssp. *lusitanicum*), 70% of Ni was in the form of free hydrated ions, 18% was complexed with carboxylic acids, mainly with citrate, whereas oxalate, malate, malonate, and aspartate complexed less than 13% of total Ni altogether, and less than 1% was bound to such amino acids as glutamic acid and glutamine [307]. In the xylem sap of *A. halleri*, 55% of Zn was complexed with citrate [308]. This is consistent with the indirect detection of Zn-OA complexes in the xylem of *A. halleri* using size exclusion chromatography coupled to inductively coupled plasma mass spectrometry (SEC-ICP-MS) [305]. At the same time, in the hyperaccumulator *S. alfredii*, 56% of Zn was transported in the hydrated form, and the rest was transported in the form of citrate and, to a lesser extent, malate [309]. Using X-ray absorption spectrometry, Salt and co-authors showed that approximately 21% of Zn in the xylem sap of *N. caerulescens* was complexed with citrate, with the remaining 79% being transported as free hydrated ions [92].

In addition to LMWOA complexes with Ni and Zn, complexes with Fe, Mg, Ca, Mn, Co, Mo, and Cd were found in the xylem sap of different plant species [119,305,310,311,312]. In the xylem sap of *P. sativum*, various complexes of Fe(II) and Fe(III) with malate and citrate were found: Fe(II)-(Malate)_2_, Fe(III)-(Malate)_2_, Fe(III)-(Citrate)_2_, as well as Fe(III)_3_-(Malate)_2_-(Citrate)_2_ and Fe(III)_3_-(Malate)_1_-(Citrate)_3_ [119]. In *S. lycopersicum*, Fe(III)_2_-(Citrate)_2_ or Fe(III)_3_-(Citrate)_3_ complexes predominated depending on the Fe:citrate ratio [120]. Complexes of Fe(III) with citrate were also found in the xylem sap of some other plant species [313], which may indicate the oxidation of Fe(II) in the xylem with subsequent reduction in the shoots upon its entry into the symplast (reviewed in [125]). The xylem sap of *P. sativum* also contained Mg/Ca/Mn/Co/Ni/Zn-(Citrate)_2_ and Mn/Ni-(Malate)_2_ complexes [119]. Citrate was directly implicated in the translocation of Ni and Cd and their accumulation in the shoots of the halophyte *Sesuvium portulacastrum*, and its content in the xylem sap and shoots increased both under separate and combined metal treatments [314]. Cadmium, at least partially, was shown to be bound to citrate in both xylem and phloem sap [315,316]. However, in the hyperaccumulators *A. halleri* and *S. alfredii*, Cd was transported in the xylem predominantly in the ionic form [183,308], though in the latter species, significant amounts of Cd-malate (22.9–24.7%) and Cd-citrate (17.6–18.9%) were also found in the xylem sap [183].

The contents of OAs in the xylem sap can change under the influence of metals. When comparing two lines of *O. sativa*, a significant increase in the contents of citrate and tartrate in the xylem sap was observed with the increase in Cd concentration, and their contents in the high-Cd-accumulating line were significantly higher than those in the normal line, indicating their contribution to higher Cd transport and accumulation in shoots [317].

In the phloem sap of the hyperaccumulator *Phyllanthus balgooyi*, Ni was complexed mainly with citrate [318], whereas in *N. caerulescens*, mainly with malate [319]. In the phloem sap of *O. sativa*, Mn and Fe were bound with citrate [316]. Due to the lower pH values of the xylem sap (pH 5–6.2) compared to the phloem sap (pH 7–8), LMWOAs, mainly citrate and malate, are primarily involved in the root-to-shoot metal transport via the xylem.

The contribution of OAs to the uptake, detoxification, and transport of metals depends on the efficiency of functioning of the transporters/channels involved in the translocation of OAs into the rhizosphere, vacuole, and conducting tissues.

## 8. Molecular Mechanisms of the Transport of LMWOAs

### 8.1. Aluminum-Activated Malate Transporters

One of the mechanisms aimed at reducing the toxic effects of metals is the activation of membrane transporters and channels involved in the secretion of OAs from the root apex into the rhizosphere. Two protein families involved in OA transport have been identified: the aluminum-activated malate transporter family (ALMT) and (some members of) the multidrug and toxic compound extrusion (MATE) family (Figure 1). The ALMTs are implicated in malate exudation, while MATE transporters release citrate into the rhizosphere to bind Al^3+^ ions [50,138,210,222,272,320,321,322,323]. In *A. thaliana*, ALMTs contribute to Al tolerance to a greater extent than MATEs [162,324]. The physiological role of ALMTs is not limited to Al tolerance, but is also associated with mineral nutrition, stomatal movement, plant–microbe interactions, fruit quality, light response, and seed development [50,325,326]. Recently, phosphorus deficiency was shown to stimulate ALMT-dependent malate secretion into the soil [327]. Therefore, the historical protein family name “ALMT” could be largely misleading.

Phylogenetic analysis showed that ALMT-like proteins are present in bacteria, Alveolata (protists, single-celled Eukaryotes), Stramenopiles (a clade of unicellular, colonial, or multicellular Eukaryotes distinguished by the presence of stiff tripartite external hairs, which includes algal protists, brown algae, oomycetes, etc.), Amoebozoa, fungi, and Viridiplantae (non-vascular and vascular green plants). Despite the fact that the family of ALMTs is quite ancient, the genes encoding these proteins are apparently absent from the animal genome [107,326].

The ALMT proteins are usually 350–500 amino-acid-long and typically have 5 to 7 putative transmembrane domains in the N-terminal half and a long hydrophilic C-terminal tail, but predictions of their secondary structure vary [50,172,173,326,328,329,330]. Transport through anion channels is driven by the membrane voltage and the chemical gradient of OAs across the plasma membrane [46,50,331]. Some ALMTs are permeable to malate anions, while others are also permeable to other organic anions (e.g., fumarate) or inorganic anions (e.g., Cl^−^, NO_3_^−^, and SO_4_^2−^) [50,326]. ALMTs vary in the degree of ion selectivity, and some of them may be involved in the transport of several LMWOAs. For example, among the seven ALMT proteins found in *Lotus japonicus*, LjALMT4 is involved in the efflux of malate, succinate, and fumarate, but not tricarboxylates, such as citrate, in nodule vasculature [332]. The permeability of ALMT6 in *A. thaliana* decreased in the following order: fumarate > malate >> citrate > Cl^−^ > NO_3_^−^ [292]. It was largely dependent on the ionic composition of permeating anions on both sides of the membrane [50].

TaALMT1, which localizes to the plasma membrane of root cells and is involved in Al-activated malate secretion in *T. aestivum*, was the first protein of the ALMT family identified, after which this family was named. The *TaALMT1* gene is constitutively expressed in the root apex and is activated by Al^3+^ ions, which leads to the release of malate, but not citrate, into the apoplast, Al binding, and reduction of its toxic effects [272,333,334]. Ryan and co-authors postulated the involvement of the TaALMT1 protein in the exudation of malate by forming an ion channel across the plasma membrane to the apoplast [328]. Transgenic *A. thaliana*, *H. vulgare*, *O. sativa*, and *N. tabacum* cell lines expressing *TaALMT1* all exhibited Al-activated malate efflux and enhanced Al tolerance [272,328,333,335,336].

Other members of the ALMT family perform a similar function in *A. thaliana* [70,161,167,337], *B. napus* [338], *B. oleracea* [172], *C. sativa* [173], *G. max* [190], *H. lanatus* [209], *Medicago sativa* [193,339], *S. cereale* [340], and *V. mungo* [200] (Table 3). However, only some of them (AtALMT1 and BnALMT1), as well as TaALMT1, are not functionally active in the absence of extracellular Al, and their activation by Al occurs within 3 to 5 min and is specific for Al [161,272,336]. In other cases, Al-induced enhancement of transport activity was observed over a longer period of time, indicating the existence of complex regulatory mechanisms. In Al-sensitive *M. sativa*, the expression of *MsALMT1* was upregulated by a 24 h treatment with 5 µM Al, which might have been due to the Al-induced upregulation of the gene encoding STOP1 transcription factor (see below) [193]. Transgenic *N. tabacum* and *A. thaliana* expressing *MsALMT1* and *GmALMT1*, respectively, showed malate efflux and higher Al tolerance compared to control plants [190,339]. Transgenic plants of *C. sativa* overexpressing *CsALMT1* were characterized by increased secretion of malate, decreased accumulation of Al, and as a consequence, a higher root elongation rate compared to the wild-type plants [173]. After 6 h of Al treatment, the level of *GmALMT1* transcript abundance was increased by more than 3-fold in the root tips of *G. max*, indicating the involvement of the plasma-membrane-located GmALMT1 in Al tolerance [190].

**Table 3 ijms-25-09542-t003:** Aluminum-activated malate-transporting proteins.

Plant Species	ALMT	Organ/Tissue	Subcellular Localization	References
*Arabidopsis thaliana*	AtALMT1	Root cells	Plasma membrane	[161,167]
	AtALMT3	Rhizodermis, especially root hair cells	Plasma membrane and small vesicles	[327]
	AtALMT4	Guard cells and mesophyll	Tonoplast	[293]
	AtALMT6	Guard cells	Tonoplast	[292]
	AtALMT9	Guard cells, mesophyll, hypocotyl, root cells, the sepals and stamens of flowers	Tonoplast	[341,342]
	AtALMT12	Guard cells	Plasma membrane	[343,344]
*Brassica napus*	BnALMT1	Root cells	Plasma membrane	[338,345]
	BnALMT2	Root cells	Plasma membrane	[338,345]
*Camelina sativa*	CsALMT1	Root cells	Plasma membrane	[173]
*Glycine max*	GmALMT1	Root cells, mainly in the root apex	Plasma membrane	[190]
*Holcus lanatus*	HlALMT1	Root and shoot cells	Plasma membrane	[209]
*Hordeum vulgare*	HvALMT1	Guard cells; root tissues; the nucellar projection, aleurone layer, and scutellum of developing grain	Plasma membrane	[346,347]
*Lupinus albus*	LaALMT1	Root cells, mainly in the apex and stele	Plasma membrane	[348]
*Oryza sativa*	OsALMT4	Root and shoot cells, mainly in the conductive tissues	Plasma membrane	[349]
*Secale cereale*	ScALMT1	Root cells, mainly in the root apex	Plasma membrane	[325,340]
*Solanum lycopersicum*	SlALMT4	Roots, leaves, flowers, and fruit	Endoplasmic reticulum	[350]
	SlALMT5	Roots, leaves, flowers, and fruit	Endoplasmic reticulum, endomembranes	[350]
*Triticum aestivum*	TaALMT1	Root cells, mainly in the apex	Plasma membrane	[272,328,333,334,351]
*Zea mays*	ZmALMT1	Throughout the plant, at a lower level in the root apex	Plasma membrane	[50,352,353]
	ZmALMT2	In mature root parts	Plasma membrane	
*Vitis vinifera*	VvALMT9	Berry mesocarp tissue	Tonoplast	[50,354]

Intraspecific differences in the secretion of LMWOAs and plant tolerance to Al may be determined by different efficiencies of ALMT-mediated machinery of OA exudation. When treated with 100 µM Al, the tolerant *T. aestivum* variety, ISOP 76, secreted more malate compared to the sensitive variety, ISOP 239. This difference between ISOP 76 and ISOP 239 can partly be related to the presence of different gene alleles, *ALMT1-2* and *ALMT1-1*, in them, respectively, resulting in different effectiveness of their responses to Al-induced stress [229]. Tolerant genotypes within a species often have significantly greater expression of corresponding *ALMT* genes than sensitive genotypes. *TaALMT1* expression in Al-tolerant genotypes of *T. aestivum* was 5- to 10-fold higher than in Al-sensitive genotypes [272,355], whereas citrate exudation in some Al-treated genotypes might even be inhibited [229,356]. In part, intraspecific differences in *T. aestivum* may be associated with genomic polymorphisms of Al-tolerance genes [357]. In *S. cereale*, malate efflux is controlled by *ScALMT* genes, which differ in copy number, expression level, and coding sequences between tolerant and sensitive genotypes [340,357].

The expression of *ALMT* genes can be induced not only by Al [161,173,338,352,358], but also by other metals, although the data available are very fragmentary. For example, the expression of *BnALMT1* and *BnALMT2* in the roots of *B. napus* was increased in response to not only Al, but also, to a lesser extent, to some other multivalent cations, such as lanthanum (La), ytterbium (Yb), and erbium (Er). Transgenic cell cultures of *N. tabacum* expressing these genes showed increased malate efflux when exposed to Al, Yb, and Er [338]. In *Stylosanthes guianensis*, the expression level of *SgALMT1* increased in response to Mn [110], whereas in *M. sacchariflorus*, the expression of *MsALMT1* was highly upregulated in response to Cd [212].

Interestingly, the toxicity of certain metals may increase along with the increase in the *ALMT* expression levels. In *O. sativa*, the plasma-membrane-located OsALMT4, one of the nine ALMTs present in this species, has been characterized in detail. The *OsALMT4* gene is expressed in roots and shoots, mainly in vascular tissues (Table 3). Transgenic *O. sativa* lines overexpressing *OsALMT4* constitutively released malate from the roots and accumulated more Mn in grain and leaf apoplast. This was accompanied by a higher concentration of malate in the xylem sap, enhanced expression of the genes encoding Mn transporters (*OsNramp5* and *OsYSL2*), as well as increased Mn sensitivity of transgenic plants compared to null plants [349].

Most members of the ALMT family, however, are not involved in Al tolerance, but perform other functions, including mineral nutrition, turgor regulation, and guard cell functioning (reviewed in [50,285,325]). For example, ZmALMT1 and ZmALMT2, whose genes are expressed in the root tissues of *Z. mays* (Table 3), are likely to assist in balancing charges during nutrient uptake [352,353]. Both proteins are localized at the plasma membrane and are involved in the transport of inorganic anions, while ZmALMT2 is also involved in the transport of organic anions, as evidenced by the secretion of malate by the roots of transgenic *A. thaliana* plants overexpressing *ZmALMT2* [353]. ZmALMT1 was shown to have higher permeability to Cl^−^ and NO_3_^−^ compared to malate. The expression of the gene encoding ZmALMT1 was at a low level in root tips, and the transport activity of ZmALMT1 was only weakly enhanced by exogenous Al in both Al-tolerant and Al-sensitive genotypes [352]. ZmALMT2 was also more permeable to inorganic ions compared to organic anions, and the *ZmALMT2* gene expression was observed in the mature root zone rather than the apex. Therefore, this protein is also associated with plant mineral nutrition and ion homeostasis rather than with Al tolerance, although *ZmALMT2* overexpression restores Al tolerance in Al-sensitive *A. thaliana* mutants [353]. It has also been postulated that in *S. lycopersicum*, SlALMT4 and SlALMT5 transport malate, while SlALMT5 also transports Cl^−^ and NO_3_^−^ [350].

The expression of *ALMT* genes is observed in different plant organs and can be organ- and tissue-specific, which is determined by the diversity of functions of ALMT family proteins (Table 3). The expression of *GmALMT1*, *ScALMT1*, and *TaALMT1* was observed predominantly in the root apex, where Al-triggered secretion of OAs usually occurs [190,272,340]. The expression of *CsALMT1* was observed only in the roots of *C. sativa* [173]. Under phosphorus deficiency, *LaALMT1* was most prominently expressed in the root apices and was also detected in the root stele of *L. albus*. It was moderately repressed by Al, suggesting a role for ALMT1 and malate mainly in metal root-to-shoot translocation [348], which is consistent with Al-induced secretion of citrate rather than malate by the roots of this species [192]. The *HvALMT1* expression was detected mainly in stomatal guard cells, in the root elongation zone, and at lateral root junctions of *H. vulgare*, independently of external Al or phosphorus supply [346]. Later, *HvALMT1* expression was also found in the nucellar projection, the aleurone layer, and the scutellum of developing grain, which is consistent with its role in anion homeostasis, distinct from Al tolerance [347]. The expression of the *SlALMT4* and *SlALMT5* genes was observed in roots, leaves, flowers, and fruit, while *SlALMT6* and *SlALMT9* were only expressed in flowers, and *SlALMT7* was expressed in flowers, roots, and immature fruit of *S. lycopersicum* [350]. The *S. officinarum* genome contains 11 ALMT genes, and in the presence of Al, a high level of transcription of some *SoALMT*s was observed in roots, though the expression of the *SoALMT1/3/6/8/10* genes was not detected there [107].

The family of *ALMT* genes is best studied in *A. thaliana*, in which it includes 14 members [50], 6 of which have been characterized in more detail (Table 3). AtALMT1 is involved in malate secretion and plays a key role in Al tolerance. The *AtALMT1* gene is expressed mainly in roots and induced by Al, and the knock-out mutants are more sensitive to Al [161,167]. AtALMT1 may also be involved in plant–microbial interactions, which confirms the diversity of its functions [326]. AtALMT3 is involved in low-phosphorus-induced malate secretion and is mainly located at the plasma membrane and in small vesicles. *AtALMT3* is significantly upregulated in phosphorus-deficient roots and is expressed in the rhizodermis, especially in the root hair cells [327]. AtALMT4 is localized at the tonoplast and is involved in the transport of malate during stomatal closure, as well as the transport of fumarate from the vacuole. The expression of the *AtALMT4* gene is observed not only in the stomatal guard cells, but also in the mesophyll cells [293]. Subcellular localization of AtALMT5 was reported to be the endoplasmic reticulum, but its physiological function has not been elucidated yet [341]. The expression of *AtALMT6*, *AtALMT9*, and *AtALMT12* was detected in the stomatal guard cells, with the first two being involved in the regulation of stomatal opening, and the last one being involved in stomatal closure [326]. AtALMT6 is located at the tonoplast and is a Ca^2+^-activated malate influx or efflux channel depending on the tonoplast potential. The *AtALMT6* gene is expressed in stomatal guard cells of leaves, stems, and flowers [292]. AtALMT9 is also located at the tonoplast of epidermal cells and was first found to mediate voltage-dependent malate fluxes directed into the vacuole [341], but more detailed characterization showed that AtALMT9 also acts as a malate-activated chloride channel [342]. *AtALMT9* expression was found in the roots and hypocotyl of young plants, stomatal guard cells, and mesophyll cells in leaves, as well as in both the sepals and stamens of flowers. Knock-out mutations led to impaired stomatal opening in the light and delayed wilting due to disrupted transport of chloride in guard cells [341,342]. In *A. thaliana*, ALMT9 is a tetramer, and the fifth putative transmembrane α-helices (TMα5) domain of each subunit contributes to the formation of the anion channel pore [329]. Unlike AtALMT6 [292], AtALMT9 is not sensitive to cytosolic Ca^2+^, though it can be activated by malate in the cytosol but not by malate in the vacuole [342]. It was assumed that AtALMT12 was located at the plasma membrane and endomembranes and was involved in the transport of Cl^−^ and NO_3_^−^, but not that of malate [344]. However, in another work [343], it was shown that AtALMT12 was located at the plasma membrane of guard cells and participated in the transport of malate from the cell, which was subsequently confirmed [359,360]. To date, the functional orthologs of *AtALMT1* have been identified, for example, in *B. napus* (*BnALMT1* and *BnALMT2*) [338], *B. oleracea* (*BoALMT1*) [172], *G. max* (*GmALMT1*) [190], *H. lanatus* (*HlALMT1*) [209], *M. sativa* (*MsALMT1*) [339], *S. cereale* (*ScALMT1*) [340], and *T. aestivum* (*TaALMT1*) [272,328]. The evolutionary relationship of ALMTs is considered in the following articles [50,107,172,341,350,361].

ALMTs are present at the plasma membrane and other biological membranes (Table 3). The proteins located at the plasma membrane, for example, AtALMT3 [327], AtALMT12 [343,344], BnALMT1 [345], BoALMT1 [172], CsALMT1 [173], GmALMT1 [190], HlALMT1 [209], OsALMT4 [349], TaALMT1 [334], ZmALMT1, and ZmALMT2 [352,353], are involved in the secretion of OAs and/or the transport of inorganic ions, whereas the proteins located at the tonoplast, such as AtALMT4 [293], AtALMT6 [292], AtALMT9 [341], SlALMT9 [362], as well as VvALMT9 in *Vitis vinifera* fruit [50,354], are involved in the transport of malate and, in the case of VvALMT9, also that of tartrate across the tonoplast, which may contribute to metal homeostasis. Both SlALMT4 and SlALMT5 in *S. lycopersicum* are located at the endoplasmic reticulum, and SlALMT5 is also located at endomembranes, providing intracellular transport of malate and inorganic ions [350].

The transport of LMWOAs across the tonoplast occurs via facilitated diffusion [58]. In addition to the involvement of AtALMT6, AtALMT9/Ma1, and SlALMT9 [168,292,329,341,342,362,363], this process is also mediated by some other mechanisms, which will be briefly discussed in the last part of this section.

Tonoplast proton pumps, such as vacuolar-type H^+^-ATPase (V-ATPase), vacuolar-type H^+^-PPase (V-PPase), and P-ATPase (PH1, PH5, etc.), drive the facilitated diffusion of malate and citrate into the vacuole [47,58,285,364,365,366]. The main function of V-ATPases is to transport protons into the vacuole by hydrolyzing ATP and to generate the pH and electrochemical potential gradients across the tonoplast to provide optimal conditions for the transport of metabolites, such as OAs [47].

The mechanism that allows the accumulation of citrate and malate in the vacuole has been described as the ‘acid trap’. At close to neutral pH values of the cytosol (7.2–7.5), malate and citrate exist mainly in the form of a dianion (malate^2−^) or a trianion (citrate^3−^), respectively (Figure 2), which can be transported into the vacuole [58,367]. Having entered the vacuole, where the pH is acidic (4.5–6), they are partially protonated (Figure 2), depending on the pH values of the vacuolar sap in a particular plant species, which maintains their electrochemical potential gradient and allows their continuous transport into the vacuole [58]. Apparently, the transport of citrate into the vacuole is easier than that of malate at their optimal concentrations in the cytosol due to more favorable thermodynamic conditions at any vacuolar pH. The accumulation of citrate in the vacuole is controlled by its concentration in the cytosol as well as by metabolic processes [58,368] and is accompanied by a significant supply of protons [369,370]. This leads to vacuolar acidification and provides a strong driving force for increased vacuolar uptake of citrate [58,369,371] mediated by V-ATPase [369,370]. The transport of citrate into the vacuole is competitively inhibited by other OAs, e.g., malate [367], which indicates the existence of similar pathways for their entry into the vacuole.

### 8.2. Multidrug and Toxic Compound Extrusion Transporters

In 2007, two scientific groups independently identified *SbMATE* in *Sorghum bicolor* [222] and *HvAACT1* (aluminum-activated citrate transporter 1) in *H. vulgare* [210] using map-based cloning. The proteins encoded by these genes belong to the MATE protein family and were shown to be involved in Al-induced citrate secretion. In both cases, Al tolerance of different genotypes of *S. bicolor* and *H. vulgare* was highly correlated with the level of expression of *MATE* genes [210,222]. Representatives of the MATE family are involved in the transport of various compounds, including anions, OAs, hormones, alkaloids, flavonoids, and anthocyanins, taking part in plant morphogenesis, ion homeostasis, detoxification of xenobiotics and metals, and tolerance to biotic stress [285,322,323,372,373,374,375].

Transporters of the MATE family are vastly distributed in all kingdoms of life and are widespread in plants [204,322,323,375]. Large number of these transporters and genes encoding them in different species emphasizes their important role in the transport of various ligands. The red algae (Rhodophyta) have 1–2 *MATE* genes, the green algae (Chlorophyta) have approximately 7–16 *MATE* genes, the bryophytes have 15–20 *MATE* genes, and *Selaginella moellendorfii* has 41 *MATE* genes [322], whereas in angiosperms, different number of *MATE* genes have been identified depending on plant species (Table 4). 

**Table 4 ijms-25-09542-t004:** Number of *MATE* genes in different species of Angiosperms.

Plant Species	Number of *MATE*s	References
*Oryza sativa*	40	[376]
	53	[377]
*Zea mays*	49	[378]
*Arabidopsis thaliana*	56	[379]
*Solanum tuberosum*	64	[56]
*Cajanus cajan*	67	[380]
*Solanum lycopersicum*	67	[381]
*Gossypium arboretum*	68	[382]
	70	[383]
*Gossypium raimondii*	70	[382]
	72	[383]
*Medicago truncatula*	70	[384]
*Populus* *trichocarpa*	71	[241]
*Linum usitassimum*	73	[385]
*Glycine* *max*	117	[386]
*Gossypium hirsutum*	128	[383]
*Nicotiana* *tabacum*	138	[375]

Phylogenetic analysis of these proteins was carried out [56,139,208,241,320,373,374,375,376,378,380,381,382,384,385,386,387,388,389,390], but only some members of the MATE family participating in LMWOA transport have been functionally characterized.

Typically, MATE transporters have 12 transmembrane domains (e.g., there are 9–12 transmembrane domains in the MATE proteins in *L. usitassimum* [385] and 14 domains in FRD3 [391]), contain about 400–700 amino acid residues, and employ conserved polar and charged Asp and Glu residues in the citrate-exuding motif (CEM) for substrate binding [56,162,164,202,205,322,323,374,375,382,388,389,390,391,392,393,394]. The conservation of these motifs and the Asp residue across evolution of land plants suggests that citrate exudation might have a critical functional relevance in their adaptation [322]. In general, MATE transporters are believed to extrude compounds by a rocker-switcher mechanism, in which the transporter exists in two conformations: straight or bent, depending on the protonation state of the acidic residues [374]. Known MATE-family citrate transporters have a characteristic cytoplasmic loop between transmembrane domains II and III [139,162,164,202,390,394]. These proteins utilize electrochemical gradients, for the maintenance of which, in most cases, H^+^ or Na^+^ ions are employed [320,322,392]. To date, citrate-permeable MATE transporters have been identified in *A. thaliana* [162], *B. oleracea* [171], *E. camaldulensis* [205], *F. esculentum* [236], *Glycine soja* [388], *H. vulgare* [210], *M. truncatula* [384], *O. sativa* [395,396], *P. trichocarpa* [241], *S. cereale* [387], *S. bicolor* [222], *T. aestivum* [227,397], *V. umbellata* [164,202], and *Z. mays* [389,393] (Table 5). The mechanism of secretion of oxalate, which, unlike citrate and malate, is considered as an end product of secondary metabolism, remains practically unexplored.

**Table 5 ijms-25-09542-t005:** Multidrug and toxic compound extrusion transporters involved in metal homeostasis.

Plant Species	MATE Transporter	Organ/Tissue	Subcellular Localization	References
*Arabidopsis thaliana*	AtMATE	Root cells	Plasma membrane	[162]
	AtFRD3	Roots, shoots, flowers, seeds, mainly in the root pericycle and conductive tissues, as well as in the veins and, to a lesser extent, in the mesophyll of the leaves	Plasma membrane	[121,313,391,398]
*Arabidopsis halleri*	AhFRD3	Roots and shoots, mainly in the root pericycle and conductive tissues, as well as in the veins and, to a lesser extent, in the mesophyll of the leaves	Plasma membrane	[398]
*Arachis hypogea*	AhFRDL1	Root cells	Plasma membrane	[139]
*Brachypodium distachyon*	BdMATE1	Root cells		[208]
	BdMATE2	Root cells		
*Brassica oleracea*	BoMATE	Mainly in root cells	Plasma membrane	[171]
*Eucalyptus camaldulensis*	EcMATE1	Root cells	Plasma membrane	[205]
*Fagopyrum esculentum*	FeMATE1	Root cells	Plasma membrane	[236]
	FeMATE2	Root and leaf cells	Golgi complex	
*Glycine max*	GmMATE13	Root apex	Plasma membrane	[390,394]
	GmMATE47	Root apex	Plasma membrane	[394]
	GmMATE75	Root cells	Plasma membrane	[399]
	GmMATE79	Root cells	Plasma membrane	
	GmMATE87	Root cells	Plasma membrane	
*Glycine soja*	GsMATE	Throughout the plant, mainly in root cells	Plasma membrane	[388]
*Hordeum vulgare*	HvAACT1	Root cells	Plasma membrane	[210,400]
*Lotus japonicus*	LjMATE1	Infection zone of nodules	Plasma membrane	[401]
*Medicago truncatula*	MtMATE66	Roots and stems	Plasma membrane	[384]
	MtMATE67	Nodules	Plasma membrane, symbiosome membrane	[402]
	MtMATE69	Roots and stems	Plasma membrane	[384]
*Oryza sativa*	OsFRDL1	Root pericycle and reproductive organs	Plasma membrane	[376,403]
	OsFRDL2	Root and leaf cells	Vesicles in cytosol	[396,403]
	OsFRDL4	All cells in the root apex	Plasma membrane	[395]
*Populus trichocarpa*	PtrMATE1	Root cells	Plasma membrane	[241]
*Secale cereale*	ScFRDL1	Throughout the plant		[387]
	ScFRDL2	Root cells		[387]
*Sorghum bicolor*	SbMATE	Rhizodermis and cortex, mainly in the root apex	Plasma membrane, tonoplast in vacuolated cells	[222,266]
*Triticum aestivum*	TaMATE1B	Root cells	Plasma membrane	[227,397]
*Vigna umbellata*	VuMATE1	Root apex	Plasma membrane	[203]
	VuMATE2	Root apex	Plasma membrane	[164]
*Zea mays*	ZmMATE1	Root cells, mainly in the apex	Plasma membrane	[393]
	ZmMATE6	Root and leaf cells		[389]

A group of MATE-family citrate transporters also facilitates Al-activated citrate exudation from the roots into the rhizosphere (Table 5), which is a critical step in the Al exclusion mechanism. For instance, the Al-tolerance genes, *AtMATE*, *BoMATE*, *EcMATE1*, *FeMATE1*, *GmMATE13/47*, *GsMATE*, *HvAACT1*, *MtMATE66*, *OsFRDL2/4*, *PtrMATE1*, *SbMATE*, *ScFRDL2*, *TaMATE1B*, *VuMATE1/2*, and *ZmMATE1*, encode plasma-membrane-localized citrate transporters that mediate citrate secretion from the root rhizodermis/cortex to the rhizosphere in Al-treated *A. thaliana* [162], *B. oleracea* [171], *E. camaldulensis* [205], *F. esculentum* [236], *G. max* [390,394], *G. soja* [388], *H. vulgare* [210,404], *M. truncatula* [384], *O. sativa* [395,396], *P. trichocarpa* [241], *S. bicolor* [107,222,266], *S. cereale* [387], *T. aestivum* [397], *V. umbellata* [164,202,203], and *Z. mays* [393], respectively (Table 5). Among them, the expression of *HvAACT1* and *TaMATE1B* was not induced by Al [210,397], but that of *AtMATE* [162], *BoMATE* [171], *EcMATE1* [205], *FeMATE1/2* [236], *GmMATE13/47* [394], *GsMATE* [388], *MtMATE66* [384], *OsFRDL2/4* [395,396], *PtrMATE1/2* [241], *SbMATE* [222,266], *ScFRDL2* [387], *VuMATE1/2* [164,202,203], and *ZmMATE1/6* [389,393] was upregulated by Al. *VuMATE1* was not expressed in the absence of Al [202], and the expression of *FeMATE1* was induced by Al only in roots [236]. The expression of *GmMATE47* could also be induced by Cd, Cu, and Hg [394], the expression of *PtrMATE1* was induced by La [241], the expression of *AtMATE* was induced by Ga [166], the expression of *GrMATE18*, *GrMATE34*, *GaMATE41*, and *GaMATE51* was induced by Cd [382], whereas the expression of *CcMATE34* and *CcMATE45* was induced by Al, Mn, and Zn [380].

The functions of MATE citrate transporters largely depend on their tissue localization (Table 5). For example, *SbMATE* is expressed in the rhizodermis and cortex, primarily in the root apices, namely, in the root distal transition zone, ensuring the secretion of citrate into the rhizosphere and the tolerance of *S. bicolor* to Al [222,266]. *S. bicolor* plants constitutively overexpressing *SbMATE* did not accumulate Al in the growing root part and had higher tolerance to Al compared to control plants [107]. Immunostaining showed that OsFRDL4 was localized in all cells in the *O. sativa* root tip [395]. Knockout of *OsFRDL2* or *OsFRDL4* resulted in decreased Al tolerance and decreased citrate secretion compared to the wild type but did not affect the internal citrate content or citrate concentration in the xylem sap, respectively [395,396]. Several MATEs that confer Al tolerance in other species are expressed not only in the root apex but throughout the roots or even in the leaves. Some examples of these include *BoMATE* [171], *FeMATE1* [236], *HvAACT1* [400], *OsFRDL2* [396,403], *TaMATE1B* [397], and *ZmMATE1/6* [389,393]. Overexpression of *BoMATE*, *GmMATE13/47*, *GsMATE*, *PtrMATE1*, *SbMATE*, and *ZmMATE6*, as well as *GhMATE1* from *G. hirsutum* in transgenic *A. thaliana* plants [171,204,222,241,388,389,394], *BdMATE* from *B. distachyon* in *Setaria viridis* [405], *HvAACT1* from *H. vulgare* in *N. tabacum* and *T. aestivum* [210,404], and *VuMATE1* from *V. umbellata* in *S. lycopersicum* [202] promoted citrate secretion and plant tolerance to Al. Three inbred lines of *Z. mays* carrying the three-copy allele of *MATE1*, originating from the regions with highly acidic soils and characterized by high *MATE1* expression, also showed increased Al tolerance [406].

It is remarkable that the same transporter, depending on its localization, can perform different functions. Among the MATE citrate transporters, HvAACT1 contributes to Al tolerance as well as long-distance Fe transport in *H. vulgare*, with the differences in the physiological function of this transporter being determined by the differences in *HvAACT1* expression in root tissues [210,400,404]. It has been shown that *HvAACT1* is expressed in the root stele, but in Al-tolerant barley accessions, a 1 kb insertion in the 5′ upstream coding region of *HvAACT1* not only enhances the level of its expression, but also alters its expression site from the stele of the mature root zone to the rhizodermis and cortex in the root tips, leading to Al-induced root citrate exudation into the rhizosphere. Evolutionary analysis showed that this insertion occurred only in some Al-tolerant cultivars as a result of adaptation to the acid soils [400]. A transposon-like element in *T. aestivum* increased *TaMATE1B* expression in the root apex, where it facilitated citrate efflux and enhanced Al tolerance [397]. The activity of transposable elements near OA transporter genes appears to have had a major impact on the evolution of Al tolerance on acid soils in many major crop species (reviewed in [330]).

Tissue localization of *MATE* gene expression can change under the influence of Al. After Al treatment, in contrast to untreated plants, *PtrMATE1* expression was observed not only in the central cylinder but was also detected in all tissues of the root apex in *P. trichocarpa* and was significantly Al-induced [241]. Aluminum treatment also extended the expression of *GmMATE13* and *GmMATE75/79* from the central cylinder to cortical and rhizodermal cells in *G. max* root tip and transgenic hairy roots, respectively, but the expression of *GmMATE87* was restricted to the central cylinder of hairy roots, irrespective of Al treatment [390,399]. Unlike that of *GmMATE75* and *GmMATE79*, the expression of *GmMATE13* was not induced by Fe deficiency, and this transporter might not be involved in the transport of Fe [390,399]. The expression of *MtMATE66* was observed not only in the rhizodermis, but also in the conducting tissues of roots and shoots, indicating its dual function: Al detoxification and Fe translocation [384]. Thus, citrate transporters located in the root stele facilitate root-to-shoot Fe translocation [313,376,387,407], whereas those located in the rhizodermal/cortical cells are involved in Al tolerance [162,266,390,395,397,399]. The localization of FeMATE2 to the *trans*-Golgi and Golgi indicates its involvement in the internal Al detoxification mechanism in the roots and leaves of *F. esculentum* [236].

One of the best studied transporters of the MATE family is FRD3 (ferric chelate reductase defective 3; Figure 1). It is located at the plasma membrane and takes part in the transport of citrate into the xylem vessels, which allows solubilizing Fe in the apoplast through chelation by citrate, thus facilitating the flow of Fe from roots to shoots as an Fe-citrate complex [313,391,407,408,409]. The *Atfrd3* mutant can transport Fe into the central cylinder of the root but is characterized by a reduced efficiency of long-distance Fe transport [313].

The expression of the *FRD3* gene was observed in both roots and shoots of *A. thaliana* and *A. halleri* (Table 5). In the roots of both species, it was found mainly in the pericycle cells and conducting tissues [391,398], whereas in the leaves, the expression was observed mainly in the veins and, to a lesser extent, in the mesophyll cells, and depended on the stage of plant development and environmental conditions [398]. The fact that *AtFRD3* is expressed in the root stele suggests that AtFRD3 is involved in citrate loading into the xylem. However, when *AtFRD3* was ectopically overexpressed in the rhizodermis/cortex of transgenic *A. thaliana* plants, it facilitated Al-activated citrate efflux to the rhizosphere and thus conferred Al tolerance [313]. The expression of *AtFRD3* in transgenic *H. vulgare* and *O. sativa* increased plant tolerance to Al by enhancing citrate efflux from the roots [324,410]. The expression of *AtFRD3* was also observed in the flowers and seeds of *A. thaliana*, and the *frd3* loss-of-function mutants were defective in early germination and were almost completely sterile. This indicates the participation of AtFRD3 in solubilizing apoplastic Fe by releasing citrate not only in the vegetative organs but also in the generative organs of the plant to ensure iron nutrition of the embryo and pollen [121].

The contribution of FRD3 to citrate secretion may be determined not only by its tissue localization, but also by the interaction between different transporters at the level of gene expression. The *GhFRD3* gene expression significantly decreased in *GhMATE1* downregulated lines of *G. hirsutum* when compared to control and Al-treated wild-type plants. These results suggest a possible interaction between *FRD3* and *MATE* expression in *G. hirsutum*. The authors concluded that under Al stress conditions, GhFRD3 plays a minimal role in Al-induced citrate secretion [204].

It is assumed that the transport of citrate and Fe with the participation of FRD3 and FPN1/IREG1 (ferroportin 1, also called iron regulated 1) can be coordinated [411]. It has been shown that FRD3 also contributes to the interaction between Fe and Zn homeostasis. A quantitative trait locus analysis identified *FRD3* as a determinant of variation in Zn tolerance among *A. thaliana* ecotypes. Higher expression of *FRD3* in response to Zn excess was co-segregating with higher Zn tolerance and reduced impact of Zn excess on Fe homeostasis [412]. The enhanced expression of the *FRD3* gene led to an increase in the contents of Fe and Zn in the endosperm of *O. sativa* grains due to an increase in the mobility of these elements throughout the plant [410]. The *A. thaliana frd3* mutant with a non-functional *FRD3* gene had lower citrate levels in the xylem, as well as lower Fe content in young and middle-aged leaves compared to wild-type plants [413], and higher Fe accumulation in the central cylinder of the root [391]. In addition, in the *A. thaliana frd3* mutant, a modification of cell walls was observed, possibly resulting in a lower degree of pectin methylesterification [411]. By binding metals, cell walls limit their entry into the cytosol, which leads to alleviation of their toxic effects [414]. Pectins with a low degree of methylation are enriched with free carboxyl groups, to which metal ions can bind. In the *Atfrd3* mutant, high expression of cell wall biosynthesis genes was also observed, including those involved in lignin biosynthesis [411], which, along with limited metal loading into the xylem, can also contribute to the accumulation of metals in the roots and their restricted entry into the shoots of this mutant.

Increased efficiency of metal loading into the xylem vessels is an important mechanism of hyperaccumulation (reviewed in [40,114,287,415]). Being involved in the loading of citrate into the xylem vessels, FRD3 may contribute to the process of hyperaccumulation. A constitutively high level of expression of the *FRD3* gene was found in the roots of the hyperaccumulators *A. halleri* and *N. caerulescens* [416,417]. Unlike *A. thaliana*, for which the possibility of direct regulation of *AtFRD3* in response to Zn was shown, the expression of the *FRD3* gene in *A. halleri* was not regulated in response to the changes in the level of Zn [398,416]. Enhanced expression of *FRD3* in the Zn-tolerant ecotype of *A. thaliana* [412], as well as in the above-mentioned hyperaccumulators [416,417], may contribute not only to the Zn tolerance [398,412], but also to the maintenance of Fe homeostasis, especially considering the higher Zn accumulation in hyperaccumulators compared to excluders [123]. In addition, this promotes root-to-shoot metal translocation in complexes with OAs, which contributes to the mechanisms of hyperaccumulation [415].

Functional homologues of FRD3 found in *O. sativa* (OsFRDL1) and *S. cereale* (ScFRDL1) are also involved in citrate loading into the xylem [376,387] (Figure 1). Immunostaining showed that ScFRDL1 was localized in all cells in the root tips and in the basal part of the root, and the expression of *ScFRDL1* was unaffected by Al treatment but upregulated by Fe deficiency [387]. Unlike *ScFRDL1*, the expression of *AtFRD3* and *OsFRDL1* was not induced by Fe deficiency [313,376]. The *Osfrdl1* knockout mutant exhibited decreased Fe content and increased Zn and Mn contents in leaves, leaf chlorosis, and lower citrate and Fe concentrations in the xylem sap [376]. As in the case of *AtFRD3*, the expression of the *OsFRDL1* gene was observed in the root pericycle, as well as in the reproductive organs, indicating their importance in seed development and distribution of Fe to the grains [121,376,403].

Although *AhFRDL1* in *Arachis hypogaea* was identified using a cloning strategy based on its sequence homology to the Fe translocation-related *MATE* genes *AtFRD3* and *OsFRDL1*, the AhFRDL1 protein sequence was more closely related to that of AtMATE, which is involved in Al exclusion in *A. thaliana* [139]. The *AhFRDL1* gene is induced by Fe deficiency and Al-imposed stress and encodes a citrate transporter involved in root-to-shoot Fe transport and Al tolerance. The expression of this gene under Fe deficiency was observed in the root stele (Table 5). In Al-treated plants, *AhFRDL1* expression was observed across the entire root tip cross-section, including the rhizodermal and cortical cells, which facilitated the entry of citrate into the rhizosphere and the detoxification of Al^3+^ ions. The overexpression of *AhFRDL1* led to the restoration of Fe transport in *Atfrd3* transgenic mutants and to Al tolerance in *AtMATE*-knockout mutants. Knocking down *AhFRDL1* in the roots caused a decrease in citrate concentration in the xylem sap and root exudates, as well as reduced the concentration of active Fe in young leaves and increased plant sensitivity to Al. The data obtained indicate that AhFRDL1 plays a significant role in Fe translocation and Al tolerance in Fe-efficient *A. hypogaea* varieties under different soil-stress conditions [139].

Transporters of the MATE family play an important role in the OA-mediated maintenance of Fe homeostasis in legume nodules, which is necessary for the process of nitrogen fixation. Unlike the above-mentioned transporters, LjMATE1 is a nodule-specific transporter that assists in the translocation of Fe from the root to nodules by providing citrate [401]. In *M. truncatula*, MtMATE67, located at the plasma membrane of nodule cells and the symbiosome membrane surrounding bacteroids in infected cells, is involved in citrate efflux from root nodule cells and maintains Fe solubility and availability for rhizobial bacteroids (Table 5). *MtMATE67* is expressed primarily in the invasion zone of mature nodules, and the loss of its function resulted in the accumulation of Fe in the apoplast of the nodules and a substantial decrease in symbiotic nitrogen fixation and plant growth [402].

Not only the dual role of some transporters but also the fine coordination between the ALMTs and MATEs contribute to the maintenance of metal homeostasis in plants.

### 8.3. Coordination between ALMTs and MATEs in Plants

Despite the existing assumption that AtALMT1-mediated malate exudation and AtMATE-mediated citrate exudation evolved independently to confer Al tolerance in *A. thaliana* [162], there are a number of mechanisms that ensure their coordinated operation. Both ALMTs and MATEs use an electrochemical gradient to transport the substrate. Therefore, the coordination of vacuolar H^+^-ATPase and plasma membrane H^+^-ATPase dictates the distribution of OAs into either the vacuolar lumen or the apoplastic space, which, in turn, determines the Al tolerance capacity in plants. The Al-induced activation of plasma membrane H^+^-ATPase was found to coincide with the secretion of OAs in the roots of *A. thaliana* [189,418], *G. max* [189,190,191], and *Vicia faba* [419], whereas in the roots of *Cucurbita pepo* [420], *H. vulgare* [421], and *T. aestivum* [228], it was inhibited by Al treatment. During Al-induced secretion of OAs into the apoplast, two tonoplast-localized isoforms of the vacuolar H^+^-ATPase subunit a (VHA-a2 and VHA-a3) were inhibited in *A. thaliana* to reduce the transport of OAs into the vacuole [418]. When the distribution of OAs into the apoplastic space was impaired, VHA-a2 and VHA-a3 were reversibly activated to drive the cytosolic OAs into the vacuole. The roots of the *vha-a2 vha-a3* mutant of *A. thaliana* accumulated less Al both in the cytosol and in the cell walls of the root apex cells due to enhanced secretion of OAs for Al detoxification and a decrease in the entry of OAs into the vacuoles of root cells. Despite the increased secretion of OAs, the contents of malate and citrate in the roots of the mutants did not differ from the wild type under the influence of Al. The *vha-a2 vha-a3* mutant displayed enhanced upregulation of the expression of *ALMT1* and *MATE* under Al stress, which was accompanied by an increase in the expression of *STOP1* and a decrease in the expression of *WRKY46* (see below). Based on these data, a model, according to which tonoplast- and plasma-membrane-localized OA transport systems function antagonistically in determining external Al exclusion or internal Al tolerance, was proposed [418]. This model shows that, in response to Al, the *AHAs*, *ALMT1*, and *MATE* transcripts are preferentially activated over the *VHA-a2* and *VHA-a3* transcripts. As a result, *A. thaliana* roots strategically suppress the vacuolar H^+^-ATPase activity to reduce the storage of OAs in the vacuoles. This, coupled with increased activity of plasma-membrane-localized OA-transporting proteins, redirects the sufficient intracellular OAs into the apoplast in order to detoxify external Al. Meanwhile, this cytosol-to-apoplast transport of OAs suppresses the expression of *VHA-a2* and *VHA-a3*. When this transport is impaired, *VHA-a2* and *VHA-a3* are reversibly activated to drive the cytosolic OAs into the vacuole, promoting internal Al tolerance [418].

### 8.4. Other Proteins Involved in the Transport of LMWOAs and Their Complexes with Metals

In addition to numerous ALMTs and MATEs, other proteins may be involved in the transport of LMWOAs across the plasma membrane. For example, in *A. thaliana*, ABCB14, belonging to the ATP-binding cassette subfamily B proteins, located at the plasma membrane of stomatal guard cells, ensures the transport of malate from the apoplast into the cells, where it acts as an osmoticum and thus modulates stomatal closure [422] (Figure 1). It was also shown that plasma membrane-localized nodulin 26-like intrinsic protein (NIP1;2) of the aquaporin family facilitates the transport of Al-malate complexes from the cell walls into the symplast of *A. thaliana* root cells with subsequent NIP1;2-mediated loading of these complexes into the xylem and their translocation to the shoots [137] (Figure 1). Obviously, NIP1;2-mediated Al removal from the cell walls of root cells requires a functional root malate exudation system mediated by AtALMT1 [137].

Compartmentation of mitochondrial metabolism requires membrane transporters that provide the mitochondrial transport of LMWOAs. Since the inner membrane of the mitochondria is impermeable to many metabolites, there are a number of specific carriers that mediate their transport between the matrix and the cytosol by the exchange mechanism, maintaining the pools of intermediates of the Krebs cycle and playing a crucial role in central carbon metabolism in plants [58,74,423]. Numerous studies conducted back in the 1970s on isolated mitochondria confirmed that exogenous malate can be transported in exchange for an inorganic phosphate by a butylmalonate-sensitive carrier into the mitochondrial matrix, where it is then rapidly oxidized by both mitochondrial MDH and NADP-ME, generating oxaloacetate and pyruvate as products [424,425,426,427,428,429].

Pyruvate transport into the mitochondria is mediated by the mitochondrial pyruvate carrier (MPC), which has been characterized in detail in *A. thaliana* [430]. The transport of other OAs across the inner membrane of the mitochondria is carried out with the involvement of the dicarboxylate/tricarboxylate carrier (DTC), dicarboxylate carrier (DIC), and succinate/fumarate carrier (SFC), which belong to the mitochondrial carrier family (MCF) and are most likely to be related to the Krebs cycle operation [49,423,431,432,433,434,435] (Figure 1). Most MCF members are relatively small, ranging from 30 to 35 kDa, around 300 amino acids in length, and have a conserved 6-transmembrane α-helical region. The carriers operate by a “ping-pong mechanism”: a substrate binds to the carrier in its c-state (cytoplasmic side open and matrix side closed state), the carrier undergoes a conformational change to a transition state and to an m-state (matrix side open and cytoplasmic side closed state), followed by the release of the substrate into the mitochondrial matrix. The counter-substrate is transported in the opposite direction by the same mechanism and then the carrier is ready to begin a new cycle (reviewed in [423]). It was suggested that DTC transports dicarboxylates, such as malate, maleate, oxaloacetate, and 2-oxoglutarate, as well as tricarboxylates, including *cis*-aconitate, citrate, isocitrate, and *trans*-aconitate, across the mitochondrial membrane via a counter-exchange mechanism [431]. These carriers mediate the flux of di- and tri-carboxylates to or from the mitochondria and have been studied in several plant species [49,423]. *DTC* is expressed in many plant tissues [431], and the expression of *CjDTC* in *Citrus junos* was induced by Al treatment [436]. In *A. thaliana*, three DIC carriers have been identified, with DIC1 and DIC2 proteins sharing 70% amino acid identity and being present in all plant organs at comparable levels, whereas DIC3 is present at low levels in flower buds and siliques [437]. In *A. thaliana*, DICs transport malate, maleate, malonate, oxaloacetate, and succinate in exchange for phosphate, sulfate, and thiosulfate at high rates, whereas 2-oxoglutarate is a very poor substrate [437]. Recently, DIC2 was shown to facilitate mitochondrial malate–citrate exchange in *A. thaliana* by functioning as a high affinity malate–citrate antiporter. In vitro and in organello analyses have demonstrated that DIC2 preferentially imports malate against citrate export [438]. DTC and DIC carriers, providing mitochondrial transport of OAs, play a significant role in a number of plant metabolic functions [49]. The mitochondrial AtSFC1 transports aconitate, citrate, and isocitrate, and, to a lesser extent, fumarate and succinate in *A. thaliana* [439]. The *AtSFC* expression was found in the root tips, hypocotyls, cotyledons, patches of veins and trichomes of mature leaves, stigmatic papillae of the carpel and anthers, as well as in the pollen grains and in-vitro-germinated pollen tubes [432]. According to existing models, the DTC, DIC, and SFC carriers may be differentially involved in the transport of OAs depending on the cyclic or non-cyclic flux mode of the Krebs cycle operation in the dark and during the day, respectively (reviewed in [423]).

Besides the involvement of AtALMT6, AtALMT9/Ma1, and SlALMT9 (see above), the vacuolar transport of LMWOAs is mediated by vacuolar transporters, ion channels, and carriers, including the tonoplast dicarboxylate transporter (tDT) [49,285,440,441] (Figure 1) and the vacuolar citrate/H^+^ symporter Cit1, which is a homologue of tDT [371]. The tDT is a 60 kDa protein that localizes to the tonoplast, contains 12 transmembrane domains, and imports and exports malate and fumarate [440], being involved in the regulation of cytosolic pH homeostasis [441]. The knock-out *AttDT* plants of *A. thaliana* accumulated less malate but more citrate in the vacuoles of leaf cells compared to the wild type [440,441]. Immunoblots using antibodies raised against Citl showed that the protein is localized to the vacuoles of juice sac cells of *C. sinensis*, mediating citrate efflux from the vacuole [371]. In *Prunus persica*, the overexpression of the *PpTST1* gene, encoding the tonoplast sugar transporter of the major facilitator superfamily (MFS), led to a decrease in the OA contents and an increase in sugar contents in fruit, suggesting its dual function in sugar accumulation and OA content reduction [442].

Plastidic transport of LMWOAs is mediated by a double-transporter system at the inner chloroplast membrane, which involves the plastidic 2-oxoglutarate/malate transporter (OMT) and the general dicarboxylate transporter (DCT; Figure 1). The former imports 2-oxoglutarate in exchange for stromal malate, while the latter exports predominantly glutamate in exchange for cytosolic malate. The transcripts of *OMT1*, *DCT1*, and *DCT2* were found in the roots, leaves, stems, siliquae, and flowers of mature *A. thaliana* plants. In leaves, the expression of these genes was induced by light, and *OMT1* was also induced by NO_3_^−^ [49,443]. Despite the fact that mitochondrial and plastid OA-transporting proteins in plants are known, their participation in plant responses to abiotic stress remains virtually unexplored.

## 9. Regulation of LMWOA Transport

Regulation of the transport of LMWOAs occurs at both the transcriptional and post-translational levels. It has been best studied for the representatives of the ALMT and MATE families and is discussed in detail in the following reviews [47,50,138,328,357,444]. Considering the limited scope of this review, we will focus only on the main aspects of the regulation of OA transport. Transcriptional regulation is carried out by transcription factors belonging to the MYB, bHLH, WRKY, and ERF families.

The MATEs and ALMT1 are positively regulated by the STOP1 system. Sensitive-to-proton rhizotoxicity 1 (STOP1) is a C2H2-type zinc finger transcription factor that co-regulates key genes involved in the mechanisms of Al tolerance in *A. thaliana* and *N. tabacum*, being required for the expression of *AtALMT1*, *AtMATE*, *NtMATE*, and some other genes associated with Al-activated malate and citrate exudation [162,165,193,445,446]. Under phosphate deficiency, Fe and Al triggered the accumulation of STOP1 in the nucleus, which, in turn, activated the expression of *ALMT1* [445]. The *stop1* mutation had no effect on *A. thaliana* sensitivity to Cd, Co, La, Mn, and sodium chloride, but it caused hypersensitivity to Al rhizotoxicity [165]. In *G. hirsutum* and *M. truncatula*, the Al-induced genes *GhALMT1*, *GhMATE1*, and *MtMATE66* were also regulated with the involvement of the STOP1 transcription system [384,447]. Moreover, overexpression of *GhSTOP1* in *A. thaliana* accelerated root growth and enhanced the expression of *AtALMT1* and *AtMATE* under Al stress conditions [447]. A complex regulatory system with minor participation of STOP1 may be involved in the regulation of the expression of *VuMATE2* and *VuMATE1*, encoding the transporters that are responsible for the early and the late phases of citrate secretion in *V. umbellata*, respectively [164,448]. Similar coordinated roles of PtrMATE1 and PtrMATE2 involved in Al-induced biphasic citrate transport have been shown in *P. trichocarpa*, but the participation of the STOP system in their regulation remains disputable [241]. A homologue of STOP1, aluminum resistance transcription factor 1 (ART1), also belongs to the family of C2H2-type zinc finger transcription factors [357]. It is localized in the nuclei of all root cells of *O. sativa* and positively regulates the expression of Al tolerance genes encoding proteins involved in the secretion of OAs, for example, *OsFRDL4* [395,449]. Increased Al-induced malate secretion in the accession of *H. lanatus* from an acid plot compared to the accession from a neutral plot was neither due to the differences in the amino acid sequence of HlALMT1, nor due to the genomic copy number difference between the two accessions. The adaptation of *H. lanatus* to acidic soils may be achieved by increasing the number of *cis*-acting elements for ART1 in the promoter region of the *HlALMT1* gene, enhancing the expression of *HlALMT1* and the secretion of malate [209]. The transcription levels of *SoSTOP1* and *SoSTAR1* (sensitive to aluminum rhizotoxicity), encoding a half-type ABC transporter [450], were dramatically increased in the roots of Al-treated *S. officinarum*, suggesting their involvement in Al tolerance pathways [107]. In the absence of STOP1, Al-induced *ALMT1* and *MATE* expression was completely suppressed in *A. thaliana* [162].

Additional transcription factors, such as calmodulin-binding transcription activator 2 (CAMTA2), positively regulate the expression of *AtALMT1* [451]. OsWRKY22, along with ART1, which regulates the expression of at least 31 downstream genes involved in Al tolerance, activates the expression of *OsFRDL4* [357,444,452]. On the contrary, AtWRKY46 acts as a negative regulator of *AtALMT1*, and the expression of *WRKY46* is inhibited by Al [453]. 

The MYB transcription factor family is one of the largest transcription factor families in plants. Fe deficiency induced the transcription factor MYB58 at the transcriptional level, and it bound to the MYB-binding site of the *FRD3* promoter, thereby acting as a direct repressor [454]. It was also demonstrated that two R2R3-MYB transcription factors, MYB1 and MYB73, influenced malate accumulation and vacuolar pH by activating vacuolar transporters in *Malus domestica* [455,456]. In particular, MdMYB73 protein bound directly to the promoters of *MdALMT9*, *MdVHA-a*, and *MdVHP1* (vacuolar pyrophosphatase 1), transcriptionally activating their expression and thereby enhancing their activities [456].

Post-translational modification, including phosphorylation or dephosphorylation and ubiquitination, also plays an essential role in the regulation of vacuolar transport of OAs [47,328]. The analysis of transgenic *Malus* x *domestica* cv. ‘Royal Gala’ showed that Ser/Thr protein kinase, salt overly sensitive 2-like 1 (SOS2L1), is required for Cd-induced phosphorylation at the Ser^358^ site of MdALMT14, enhancing its stability, and indicated the important role of phosphorylation in the regulation of OA secretion under metal-induced stress [358]. The activity of AtALMT12 (QUAC1 anion channel) is regulated via phosphorylation by the open stomata 1 (OST1) kinase, which is activated by abscisic acid (ABA) under stress conditions [360].

The hormonal regulation of OA secretion has not been sufficiently studied. It was shown that the expression of the *AtALMT1* gene was induced by indole-3-acetic acid (IAA) and ABA, while it was not induced by methyl jasmonate or salicylic acid [337]. Similarly, IAA stimulated *GmMATE* expression in response to Al, which was accompanied by an increase in citrate secretion by *G. max* roots [191]. Exogenous co-treatment with Al and IAA led to an increase in the secretion of malate by *T. aestivum* roots and a decrease in the accumulation of Al in the root apex compared to plants treated solely with Al. Aluminum treatment increased the accumulation of IAA, which, along with the high correlation between malate secretion and endogenous IAA levels, confirmed that IAA is involved in the Al-induced efflux of malate in *T. aestivum* [228,457]. It was recently shown that the plasma membrane MATE transporter DTX30, localized in the root rhizodermis, modulates auxin levels in the root to regulate cell elongation and promotes citrate exudation to alleviate Al toxicity [458]. Aluminum-induced malate efflux is negatively regulated by ethylene by targeting TaALMT1-mediated malate efflux via an unknown mechanism [459]. In contrast, the expression of *AtFRD3* and many other important Fe-regulated genes is upregulated in response to treatment with ethylene, and thus, AtFRD3 might be acting as a link between hormonal regulation and metal homeostasis [460]. ABA was shown to induce the anion channel AtALMT4, which mediates malate efflux from the vacuole during stomatal closure and whose activity is controlled by phosphorylation of C-terminal serine [293].

Polyamines are known as protective molecules and, along with phytohormones, can be involved in a complex signaling system, playing a significant role in the regulation of plant stress tolerance [461]. Treatment with Al triggered an enhancement in the endogenous level of free putrescine, which induced the synthesis of NO, which, in turn, stimulated citrate secretion by *P. vulgaris* roots in response to Al [462]. 

Gaseous signaling molecules H_2_S and NO are involved in the regulation of expression of *MATE* genes. Aluminum treatment stimulated nitrate-reductase-dependent NO production in *G. max* root apices, which, in turn, induced H_2_S accumulation by regulating the key enzymes involved in H_2_S metabolism. Subsequently, H_2_S upregulated both the plasma membrane H^+^-ATPase pumping H^+^ outside the cell to create an electrochemical gradient, thus providing a driving force for the activation of secondary transporters, and the expression of *GmMATE13/47*. As a result, Al-induced citrate secretion increased, and Al accumulation in the root tips decreased [394]. However, the use of exogenous H_2_S did not have a significant effect on the secretion of malate in *T. aestivum* or oxalate in *F. esculentum*, indicating the possible specificity of this regulatory mechanism for Al-stressed citrate-secreting plants [394].

Low pH values can affect the expression level of *ALMT* genes mediating OA secretion by the roots. The low pH values and hydrogen peroxide were found to upregulate *ALMT1* transcripts [167,337,445], which is consistent with Al-induced peroxide accumulation in the root tips and a decrease in the cellular pH [337]. In addition, at low pH values in the rhizosphere, the amounts of bioavailable forms of metals increase and, consequently, their toxic effects are manifested. In *G. max*, the expression of the *GmALMT1* gene was coordinately regulated by low pH, Al, and phosphorus, which indicates the possible role of phosphorus as a signaling molecule [190]. Interestingly, gamma-aminobutyric acid (GABA), which accumulates in plant tissues in response to biotic and abiotic stress, was shown to act as a negative regulator of *TaALMT1*, resulting in altered root growth and tolerance to Al, acid pH, and alkaline pH [351]. ALMT proteins contain identified GABA-binding sites required for GABA regulation [50], and cytosolic GABA inhibits anion transport by TaALMT1 [463].

Thus, LMWOAs play an important role in plant responses to metal-induced stress. In addition, these small organic molecules are involved in the transport of metals. Altogether, this is aimed at maintaining metal homeostasis. Therefore, studying the role of LMWOAs in the mechanisms of plant tolerance to metals is one of the promising directions for further research.

## 10. Conclusions and Outlook

Further studies of LMWOAs are of theoretical and practical significance. Although the important role of LMWOAs in metal homeostasis has been established, many problems that require the researchers’ attention still remain unsolved. Having entered the cytosol, metals can form complexes with available free ligands, among which, besides LMWOAs, histidine, nicotianamine, phytochelatins, and metallothioneins play an important role (reviewed in [8,40,41,42,115,295,464]). Despite the high probability of the formation of metal complexes with metal-binding ligands in the cytosol, no clear evidence of that is available yet. This is due to the facts that (i) no cytoplasm isolation techniques retaining the in vivo metal microenvironment have been developed so far, and (ii) the metal concentration in the cytoplasm is generally below the detection limit of techniques that can assess metal speciation. The situation is also complicated by the fact that LMWOAs can form complexes of different structures [118,119].

Essential transition metals may be present in nanomolar to micromolar concentrations in the cytoplasm [465], while the concentration of LMWOAs is much higher [88,96]. Since a number of ligands may be present in the cytosol in different ratios, there may be competition both between ligands for one metal, and between metals for one ligand. The ratio between different types of ligands may vary for different tissues as well, which may result from the heterogeneity of localization of corresponding transporters, as well as the uneven patterns of metal distribution over tissue or cell types [40,414]. However, due to the obvious difficulties in visualizing ligands and their complexes with metals in planta, such works are still rare [466].

Both deficiency and excess of various metals can, to a varying degree, affect the synthesis of low-molecular-weight ligands in plant cells, and this effect may differ for different ligands, which, consequently, will lead to a change in the buffering capacity of the cytosol [40]. The data on the combined effects of different metals on OA-mediated machinery are scarce [153,219,314], which, in connection with the problem of polymetallic stress, makes this area of research very relevant. The study of additive, synergistic, or antagonistic effects under the combined action of various stress factors is also promising and of practical importance. For example, Al toxicity and phosphorus deficiency as a result of poor bioavailability of phosphate commonly coexist on acid soils [50]. In *G. max,* phosphorus deficiency mainly induced oxalate and malate exudation, Al treatment induced citrate exudation, whereas under combined effects, the secretion of OAs decreased [66]. The carboxylates chelate Al^3+^ ions and set phosphates free. However, during the secretion of OAs, a large amount of carbon is lost, and plants have to maintain a balance between the positive effects of OA release and the disadvantages of losing valuable carbon sources, which is achieved by fine-tuning the regulatory mechanisms of OA secretion and synthesis, which requires further studies. Despite significant progress in understanding the role of LMWOAs in maintaining metal homeostasis, the detailed relationships between various external and internal OA-mediated metal detoxification pathways remain to be deciphered.

It is widely known that nutritional deficiency is a limiting factor that reduces crop productivity and yield and is a widespread cause of human diseases and child mortality [467,468,469]. One of the techniques employed to solve this problem is biofortification, which involves the development of approaches to increase the level of bioavailable elements, primarily that of Fe and Zn, in crops, mainly in various Poaceae species [468,470]. Since metal homeostasis depends on close coordination between low-molecular-weight ligands and membrane transporters [39], the former play an important role in the biofortification of Fe and Zn [471]. Manipulation of the OA pathways as a biotechnological tool is described in detail in the following review [46]. For example, in transgenic *O. sativa* plants expressing *AtFRD3* in combination with *AtNAS1* (encodes the key enzyme of nicotianamine biosynthesis, nicotianamine synthase) and *PvFER* (encodes storage protein ferritin), or with *PvFER* alone, the concentrations of citrate and Fe in the xylem sap increased twice compared to the control, and the contents of Fe and Zn in polished and unpolished grains also significantly increased. In addition, the transformed lines were more tolerant to Fe deficiency and Al toxicity [410]. Overexpression of OA efflux transporters in transgenic plants also improved their ability to cope with biotic and abiotic stress. A combined approach to improve both the production and secretion of OAs may be a useful strategy for increasing plant metal tolerance. The use of transgenic plants with enhanced synthesis and exudation of OAs could help to improve metal tolerance and reduce the use of fertilizers [46]. However, it must be taken into account that practical use of transgenic plants in agriculture may be limited by the existing risks of the use of genetically modified organisms [472,473].

Future research should be focused on identifying different genes and pathways for the biosynthesis and transport of OAs that are modified in the plants to secrete more secondary metabolites, protecting them against metal toxicity when grown on metal-contaminated soils. Among the promising approaches aimed at increasing the secretion of OAs by plants there are the intercropping, i.e., co-cultivation of certain plant species, and the inoculation with rhizobacteria. Intercropping of hyperaccumulators and crops is being actively studied [168,176,191,474]. When *V. faba* and the hyperaccumulator *Arabis alpina* were co-cultivated, the total amounts of OAs secreted by the roots were 578.8% and 37.8% greater than those secreted by mono-cropped plants, respectively. Additionally, intercropping enhanced the phytoremediation potential of these plant species toward Pb- and Cd-contaminated soils [168]. When *Z. mays* and the Cd hyperaccumulator *Sonchus asper* were grown together, root exudation of citrate and oxalate affected the bioavailability of Cd in the soil, which increased Cd uptake and accumulation in *S. asper* but inhibited Cd accumulation in *Z. mays* [474]. The co-inoculation of *Z. mays* with the beneficial rhizobacteria *Pantoea* sp. strain WP-5 and biogas residues significantly elevated the concentrations of acetate and citrate in the root exudates of *Z. mays*, which promoted the phytostabilization and accumulation of Cd in the roots and restricted its translocation to the shoots [475].

Exogenous plant treatment with various LMWOAs also affects the bioavailability of metals, their accumulation in plants, and the manifestation of metal toxic effects, although the results obtained in such experiments are not always consistent (reviewed in [45,476]). For example, the treatment of *Brassica juncea* with citrate and oxalate alone or in combination with EDTA increased the bioavailability of Cd and Zn in the soil [477]. Treatment with citrate significantly enhanced Cd uptake and accumulation in the roots and shoots of *B. napus* and *Bidens pilosa* and alleviated Cd toxicity [478,479]. Exogenous treatment with citrate also significantly increased the contents of Mn, Fe, Co, Ni, and Cr in the leaves of *N. caerulescens* and *T. arvense* [480]. Combined exposure of *Mentha piperita* to Ni (100, 250, or 500 μM) and citrate (5 mM) increased the Ni content in the roots, stems, and leaves compared to the plants growing only in the presence of Ni [481]. Furthermore, OAs stimulated uranium accumulation in *B. juncea* and *B. chinensis* [482] and increased Cr accumulation in *S. lycopersicum* [483]. At the same time, exogenous treatment of *O. sativa* with citrate and malate decreased the accumulation of Cd in the leaves due to the upregulation of *OsHMA3* encoding the tonoplast-localized heavy metal ATPase, which is involved in the vacuolar sequestration of Cd in roots [484], which confirmed the possibility of obtaining divergent results on the effects of exogenous LMWOAs on the accumulation of metals in different plant species. However, the use of some chemicals, including LMWOAs, may play a significant role in bio- and phytoremediation of metals or in alleviating metal-induced toxicity [89,183,298,479,485,486,487,488,489,490,491,492,493], although it requires an assessment of its economic feasibility. Nevertheless, despite the existing shortcomings, the development of various technologies for biofortification, phytoremediation, and phytomining, including those not employing transgenic plants, is an important direction for future research, in which significant attention should be paid to the studies of LMWOAs.

## Figures and Tables

**Figure 1 ijms-25-09542-f001:**
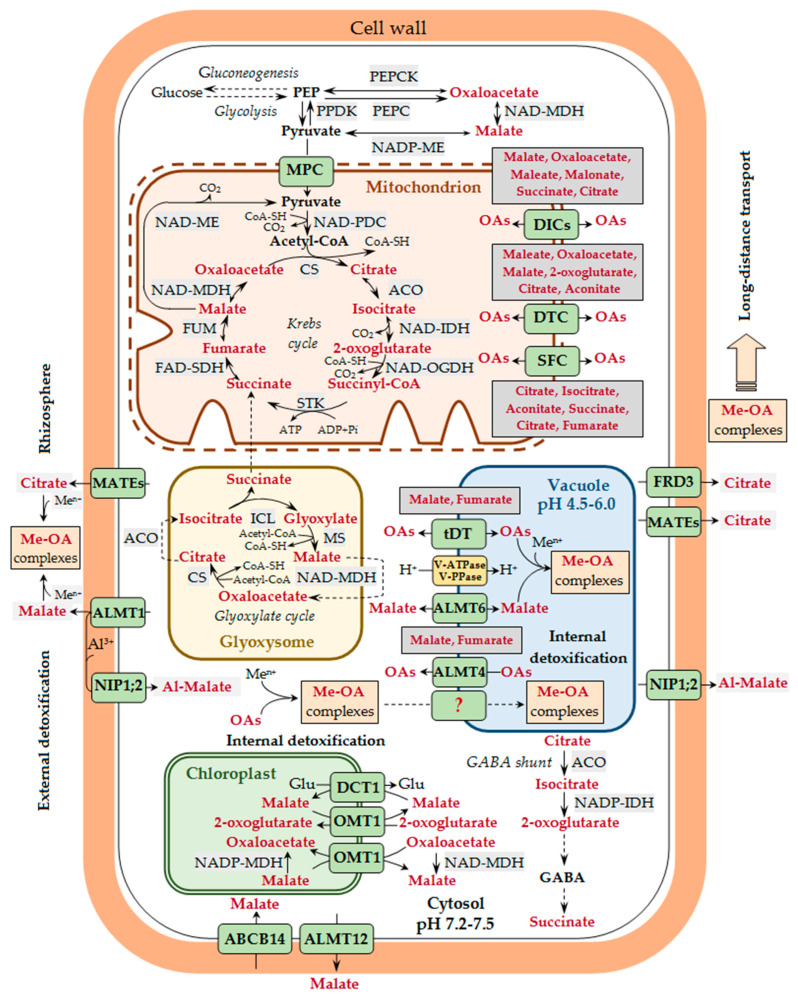
Participation of organic acids in metal transport and detoxification in plants. A generalized scheme is presented, without taking tissue specificity into account. The release of root exudates containing various organic acids (OAs) is the basis of the exclusion tolerance mechanism. Malate is secreted by aluminum-activated malate transporter 1 (ALMT1), whereas citrate is secreted by some members of the multidrug and toxic compound extrusion transporter (MATE) family (e.g., AtMATE, BoMATE, EcMATE1, FeMATE1, GmMATE13/47, GsMATE, HvAACT1, MtMATE66, OsFRDL2/4, PtrMATE1, SbMATE, ScFRDL2, TaMATE1B, VuMATE1/2, and ZmMATE1), which are located at the plasma membrane of the root rhizodermal cells. In the rhizosphere, OAs bind metal ions (Me^n+^) with the formation of complexes of various structures (Me-OA complexes), which affects metal entry into the plant. Aluminum ions (Al^3+^) in the cell walls of root rhizodermal cells may bind to malate, which is transported there by ALMT1, and the resulting complexes (Al-malate) are transported across the plasma membrane into the cytosol with the involvement of nodulin 26-like intrinsic protein (NIP1;2). In the cell, the major site of OA biosynthesis is the mitochondria and the glyoxisomes, where the reactions of the Krebs cycle and the glyoxylate cycle, respectively, take place. Pyruvate transport into the mitochondria is mediated by the mitochondrial pyruvate carrier (MPC). The transport of other OAs across the inner membrane of the mitochondria is carried out by the exchange mechanism with the involvement of dicarboxylate carriers (DICs), dicarboxylate/tricarboxylate carrier (DTC), and succinate/fumarate carrier (SFC). Having entered the cytosol via the plasma membrane transporters, metal ions bind to different low-molecular-weight ligands, including OAs, though the stability of Me-OA complexes at neutral pH values is lower than that of metal complexes with nicotianamine and histidine. The possibility of translocation of Me-OA complexes across the tonoplast cannot be excluded, though the mechanism of such transport is unknown yet (this pathway is designated by a dotted line and the unknown transporter by a question mark). The entry of metal ions into the vacuole is carried out by different vacuolar transporters, whereas the transport of OAs across the tonoplast is mediated by the tonoplast dicarboxylate transporter (tDT) and the vacuolar citrate/H+ symporter Cit1 (the latter is not shown in the figure). In *A. thaliana*, OA transport across the tonoplast is also carried out by ALMT4 and ALMT6, whose gene expression was shown in stomatal guard cells, and for ALMT4—also in leaf mesophyll. Potential substrates for the mitochondrial carriers (DICs, DTC, and SFC) and the vacuolar OA-transporting proteins (tDT and ALMT4) are shown next to the corresponding transporters/carriers/channels. Metal binding to OAs in the vacuole is an important internal metal detoxification mechanism. The transport of malate across the plasma membrane in stomatal guard cells in *A. thaliana* may be carried out by ALMT12 and ABCB14 (ATP-binding cassette subfamily B protein), being directly involved in the stomatal movement and indirectly involved in maintaining metal homeostasis. Metals are translocated from roots to shoots via the xylem mainly as complexes with OAs. Citrate is transported into the xylem vessels by the FRD3 transporter (ferric chelate reductase defective 3) located at the plasma membrane of root central cylinder cells, as well as other proteins of the MATE family (e.g., OsFRDL1, PtrMATE1, MtMATE66, and ScFRDL1). Plastidic transport of OAs in the cells of photosynthetic tissues is carried out via a double-transporter system at the inner chloroplast membrane involving the plastidic 2-oxoglutarate/malate transporter (OMT1) and the general dicarboxylate transporter (DCT1). The arrows show the direction of transport. Designations: Acetyl-CoA, acetyl coenzyme A; ACO, aconitase; CoA-SH, coenzyme A; CS, citrate synthase; FAD-SDH, FAD-succinic dehydrogenase; FUM, fumarase; GABA, gamma-aminobutyric acid; Glu, glutamic acid; ICL, isocitrate lyase; MS, malate synthase; NAD-IDH, NAD-isocitrate dehydrogenase; NAD-MDH, NAD-malate dehydrogenase; NAD-ME, NAD-malic enzyme; NAD-OGDH, NAD-2-oxoglutarate dehydrogenase; NADP-ME, NADP-malic enzyme; NADP-IDH, NADP-isocitrate dehydrogenase; PDC, pyruvate dehydrogenase complex; PEP, phosphoenolpyruvate; PEPC, phosphoenolpyruvate carboxylase; PEPCK, phosphoenolpyruvate carboxykinase; PPDK, pyruvate orthophosphate dikinase; STK, succinate thiokinase (succinyl-CoA synthetase).

**Figure 2 ijms-25-09542-f002:**
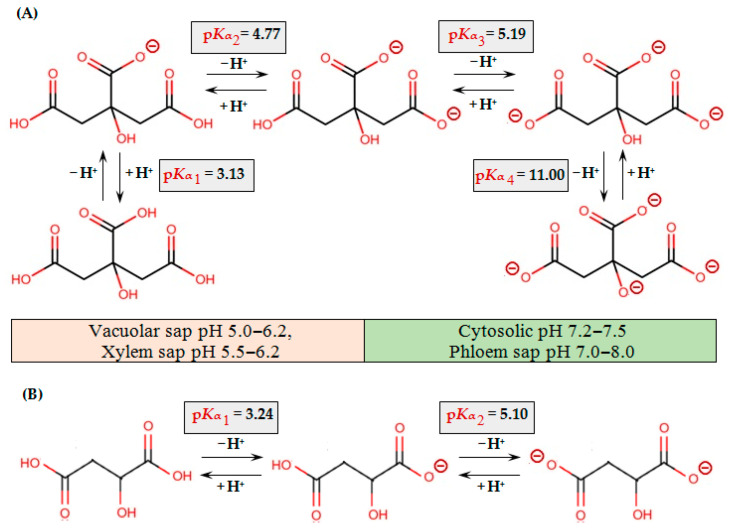
Protonation/deprotonation equilibria of citric (**A**) and malic (**B**) acids. p*K*_α_ values for citric and malic acids are presented according to [116,117], respectively.

**Table 1 ijms-25-09542-t001:** p*K*_α_ values for some LMWOAs at an ionic strength of 0 (according to [116,117,131]).

Carboxylic Acid	p*K*_α1_	p*K*_α2_
Acetic	4.76	-
Aconitic	2.62	3.97
Citric	3.13	4.77
Fumaric	3.02	4.48
Maleic	1.92	6.27
Malic	3.24	5.10
Malonic	2.85	5.69
Oxalic	1.25	4.26
Succinic	4.20	5.36
Tartaric	3.04	4.37

## Data Availability

Not applicable.

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
