# Peer review of "The Role of Low-Molecular-Weight Organic Acids in Metal Homeostasis in Plants"

_ijms, 2024, doi:10.3390/ijms25179542_

Round 1

Reviewer 1 Report

Comments and Suggestions for Authors

Dear Authors,

This review manuscript, titled "The Role of Low-Molecular-Weight Organic Acids in Metal Homeostasis in Plants," investigates the crucial role of low-molecular-weight organic acids (LMWOAs) in plant metal management. LMWOAs act as essential metal-binding ligands, playing a key part in maintaining metal homeostasis, various metabolic processes, and plant responses to stress. The review focuses on understanding LMWOA accumulation and function in two contrasting plant strategies: excluders (storing metals primarily in roots) and hyperaccumulators (concentrating metals in shoots).

This manuscript presents a well-organized and insightful investigation into the role of LMWOAs in plant metal homeostasis. I strongly recommend its publication without revisions.

Congratulations to the authors for their commendable work on this subject.

Comments on the Quality of English Language

Author Response

Comments: 

This review manuscript, titled "The Role of Low-Molecular-Weight Organic Acids in Metal Homeostasis in Plants," investigates the crucial role of low-molecular-weight organic acids (LMWOAs) in plant metal management. LMWOAs act as essential metal-binding ligands, playing a key part in maintaining metal homeostasis, various metabolic processes, and plant responses to stress. The review focuses on understanding LMWOA accumulation and function in two contrasting plant strategies: excluders (storing metals primarily in roots) and hyperaccumulators (concentrating metals in shoots).

This manuscript presents a well-organized and insightful investigation into the role of LMWOAs in plant metal homeostasis. I strongly recommend its publication without revisions.

Congratulations to the authors for their commendable work on this subject.

Response:  We deeply thank the Reviewer for appreciating our work!

Reviewer 2 Report

Comments and Suggestions for Authors

The review extensively covers the sources of low-molecular-weight organic acids (LMWOAs) in soil, including root exudates, plant residues, and microbial metabolites. It details the types of LMWOAs providing a thorough understanding of their origins and prevalence in different ecosystems. This comprehensive view describing the complex composiiton of LMWOAs and root exudates etc  enhances understanding of how plants interact with their environment through exudation. The review also acknowledges the variability in LMWOA composition due to factors such as plant species, soil type, and nutrient availability. This recognition of variability adds depth to the understanding of soil-plant interactions and adaptation mechanisms.

However, The review covers a broad range of topics (seems a book chapter rather than a review), which enhances general understanding but may sacrifice depth in specific areas. A more focused approach should provide more detailed mechanistic insights

The main limitation of the present review lies for me in the multitude of topics and areas of reseach they try to cover —with some being more extensively discussed others too little —, which is almost impossible to fit into one review. The review is more a loose collection of various topics with only limited depth for each section/topic, scope of the review not clear - it would be advisable to focus only on a few areas and extract information better for drawing conclusions and providing novel directions for LMWOA. 

In addition, the manuscript is too much wording and difficult to follow for readers. In fact, it includes more than 1800 lines (and ca. 500 citations) for the main text, although just two figures are shown. The authors need to consider how to gain higher visibility in the academic societies and how to attract more readers.

Additionally, Authors should provide qualitative descriptions (i.e. quantitative data on exact concentrations or specific effects across all mentioned species and conditions)  of LMWOA composition and changes under stresses, it lacks. This could limit the ability to compare findings across studies rigorously.

Several sentances/passages mention studies without direct citation or specific references, potentially reducing the ability to verify claims or replicate findings easily. Clearer citation practices would enhance the review's credibility and facilitate further research.

 The reference list must be streamed  and focuss-oriented! 

Comments on the Quality of English Language

Minor editing of English language required

Author Response

Comment 1: The review extensively covers the sources of low-molecular-weight organic acids (LMWOAs) in soil, including root exudates, plant residues, and microbial metabolites. It details the types of LMWOAs providing a thorough understanding of their origins and prevalence in different ecosystems. This comprehensive view describing the complex composiiton of LMWOAs and root exudates etc  enhances understanding of how plants interact with their environment through exudation. The review also acknowledges the variability in LMWOA composition due to factors such as plant species, soil type, and nutrient availability. This recognition of variability adds depth to the understanding of soil-plant interactions and adaptation mechanisms.

 However, The review covers a broad range of topics (seems a book chapter rather than a review), which enhances general understanding but may sacrifice depth in specific areas. A more focused approach should provide more detailed mechanistic insights

The main limitation of the present review lies for me in the multitude of topics and areas of reseach they try to cover —with some being more extensively discussed others too little —, which is almost impossible to fit into one review. The review is more a loose collection of various topics with only limited depth for each section/topic, scope of the review not clear - it would be advisable to focus only on a few areas and extract information better for drawing conclusions and providing novel directions for LMWOA. 

Response 1: We thank the Reviewer for the valuable recommendations helping to improve the quality of our manuscript.

The aim of our review is indeed to give a general overview of the topic to provide a general understanding of the role of organic acids in metal transport and homeostasis in plants. Some sections, for example, the biosynthesis of LMWOAs , the sources of LMWOAs in soil, the functions of root exudates or the regulation of transport of LMWOAs, have been covered in detail in other reviews [López-Bucio et al., 2000; Piechulla, Heldt, 2005; Etienne et al., 2013; Maurino et al., 2015; Igamberdiev, Eprintsev, 2016], [Sokolova, 2020], [Bais et al., 2006; Van Dam, Bouwmeester, 2016; Panchal et al., 2021; Ahlawat et al., 2024] and [Ryan et al., 2011; Delhaize et al., 2012; Liu et al., 2014; Sharma et al., 2016a; Yang et al., 2019; Huang et al., 2021a], respectively, so these sections have not been covered in detail in our manuscript, but only the main aspects were discussed. Still, they are important to mention due to the logic of our review, so we have included these sections and provided the references for the corresponding reviews. Thus, due to space limitations, we focused more on the topics not covered in other reviews, e.g. the comparison of the role of OAs in metal homeostasis in hyperaccumulating and non-accumulating species.

Comment 2:  In addition, the manuscript is too much wording and difficult to follow for readers. In fact, it includes more than 1800 lines (and ca. 500 citations) for the main text, although just two figures are shown. The authors need to consider how to gain higher visibility in the academic societies and how to attract more readers.

Response 2: Thank you. We have shortened the manuscript (now there are 46 pages of text instead of 51). As to the illustrative material, besides the two figures, the data have been summarized in 5 tables. Moreover, the first figure is a comprehensive scheme depicting the participation of organic acids in metal transport and detoxification in plants, as well as OA biosynthesis and transport.

Comment 3: Additionally, Authors should provide qualitative descriptions (i.e. quantitative data on exact concentrations or specific effects across all mentioned species and conditions)  of LMWOA composition and changes under stresses, it lacks. This could limit the ability to compare findings across studies rigorously.

Response 3: Thank you. In the majority of works, due to technical limitations or specific aims of the study, only a few OAs were quantified, whereas the whole spectrum of the OAs secreted or accumulated was covered only in a few works. In many works, specific OAs were quantified at different time points and in different conditions (e.g. different metal concentrations used, different growth conditions or sampling conditions), which precludes from direct comparing the data. Moreover, giving quantitative data for all species and conditions mentioned would significantly enlarge the review, while the Reviewer already finds the manuscript too big. Thus, we believe that based on the data summarized in the provided tables and text of the review, the readers will easily find the quantitative data  in the original papers for the species of their interest.

Comment 4: Several sentances/passages mention studies without direct citation or specific references, potentially reducing the ability to verify claims or replicate findings easily. Clearer citation practices would enhance the review's credibility and facilitate further research.

Response 4: Thank you. We have carefully checked the text and added the missing references.

 Comment 5: The reference list must be streamed  and focuss-oriented! 

Response 5: Thank you. The reference list has been formatted according to the rules of IJMS. The references have been carefully chosen to fit the topic of the review.

Comment 6: Minor editing of English language required

Response 6:  Thank you, we have checked the usage of the English language.